# Quantitative proteomics and single-nucleus transcriptomics of the sinus node elucidates the foundation of cardiac pacemaking

Nora Linscheid [1,8], Sunil Jit R.J. Logantha [2,8], Pi Camilla Poulsen[1], Shanzhuo Zhang[3], Maren Schrölkamp[1], Kristoffer Lihme Egerod[4], Jonatan James Thompson[4], Ashraf Kitmitto[2], Gina Galli [2], Martin J. Humphries [5], Henggui Zhang[6], Tune H. Pers[4], Jesper Velgaard Olsen [7], Mark Boyett[2] & Alicia Lundby[1,7]

The sinus node is a collection of highly specialised cells constituting the heart's pacemaker. The molecular underpinnings of its pacemaking abilities are debated. Using high-resolution mass spectrometry, we here quantify >7,000 proteins from sinus node and neighbouring atrial muscle. Abundances of 575 proteins differ between the two tissues. By performing single-nucleus RNA sequencing of sinus node biopsies, we attribute measured protein abundances to specific cell types. The data reveal significant differences in ion channels responsible for the membrane clock, but not in $Ca^{2+}$ clock proteins, suggesting that the membrane clock underpins pacemaking. Consistently, incorporation of ion channel expression differences into a biophysically-detailed atrial action potential model result in pacemaking and a sinus node-like action potential. Combining our quantitative proteomics data with computational modeling, we estimate ion channel copy numbers for sinus node myocytes. Our findings provide detailed insights into the unique molecular make-up of the cardiac pacemaker.

[1] Department of Biomedical Sciences, Faculty of Health and Medical Sciences, University of Copenhagen, København 2200, Denmark. [2] Division of Cardiovascular Sciences, University of Manchester, Manchester M13 9NT, UK. [3] School of Computer Science and Technology, Harbin Institute of Technology, Haerbin Shi 150006, China. [4] Novo Nordisk Foundation Center for Basic Metabolic Research, Faculty of Health and Medical Sciences, University of Copenhagen, København 2200, Denmark. [5] Wellcome Trust Centre for Cell-Matrix Research, Faculty of Biology, Medicine & Health, University of Manchester, Manchester M13 9PT, UK. [6] Biological Physics Group, School of Physics & Astronomy, University of Manchester, Manchester M13 9PL, UK. [7] The Novo Nordisk Foundation Center for Protein Research, Faculty of Health and Medical Sciences, University of Copenhagen, København 2200, Denmark. [8] These authors contributed equally: Nora Linscheid, Sunil Jit R. J. Logantha. Correspondence and requests for materials should be addressed to M.B. (email: mark.boyett@manchester.ac.uk) or to A.L. (email: alicia.lundby@sund.ku.dk)

The rhythmic contraction of the heart is initiated by electrical impulses (action potentials) generated in the the sinus node, a small population of highly specialised cells located in the dorsal wall of the right atrium[1]. Myocytes of the sinus node are known to have the ability to generate cyclical action potentials, each of which initiates a heartbeat, thereby enabling the sinus node to act as the heart's pacemaker. However, less is known about what makes this cluster of cells unique. Specifically, which proteins enable the sinus node to generate repetitive action potentials thereby ensuring the foundation for the rhythmic contraction of the heart? Pacemaking is suggested to be the result of a membrane clock or a $Ca^{2+}$ clock and the relative importance of the two is keenly debated[2]. During the pacemaker potential (the automatic depolarisation at the end of an action potential) characteristic of sinus node myocytes, the membrane and $Ca^{2+}$ clocks both generate inward currents driving the membrane potential to the threshold for initiation of the next action potential. The membrane clock comprises an ensemble of plasma membrane ion channels (such as the funny channel and L- and T-type $Ca^{2+}$ channels) carrying inward currents, and other ion channels ($K^+$ channels) carrying opposing outward currents. Unlike the membrane clock, the $Ca^{2+}$ clock is driven by an intracellular signal. The $Ca^{2+}$ clock involves localised spontaneous $Ca^{2+}$ releases from the sarcoplasmic reticulum (SR). The $Ca^{2+}$ releases occur via the ryanodine receptor (and involve other $Ca^{2+}$ clock proteins such as Serca2) and ultimately result in $Ca^{2+}$ extrusion via the $Na^+$-$Ca^{2+}$ exchanger and, therefore, a net inward current (the exchanger is electrogenic).

To distinguish between the two competing explanations of pacemaking, we set out to investigate the protein composition that endows the sinus node with its unique characteristics. By means of high-resolution mass spectrometry, we quantify protein abundances across sinus node and neighbouring, non-pacemaking atrial muscle, enabling us to determine which proteins are differentially expressed. Our data support the membrane clock as the key driver of cardiac pacemaking. The quantitative proteomic atlas of the sinus node we report here offers insights into the biology of the cardiac pacemaker that goes beyond the membrane and $Ca^{2+}$ clocks. We present high-resolution proteomics data for 7248 proteins isolated from the mouse sinus node and the atrial muscle abutting it. By performing single-nucleus RNA sequencing (snRNA-seq) experiments of sinus node biopsies we furthermore outline in which particular cell type the quantified proteins are expressed. That is, by presenting single-cell transcriptomics data of the sinus node we add information on cell type-specificity to our findings. For instance, we find that membrane clock proteins are essentially exclusively expressed in sinus node myocytes. With this knowledge, we use our proteomics data combined with computational modeling to calculate absolute copy numbers of voltage-gated ion channels per sinus node myocyte. The information contained in this data forms a foundation to evaluate sinus node function as well as molecular changes underlying pathological states of the sinus node.

## Results

**Deep proteomes of sinus node and atrial muscle biopsies**. To define protein expression in the sinus node, sinus node and adjacent right atrial muscle were dissected from mouse hearts and analysed by high-resolution mass spectrometry (Fig. 1a). Owing to the small biopsy size, samples were dissected from 30 mice in total, and biopsies from ten animals were pooled, resulting in three biological replicates for each tissue type. The localisation and boundary of the sinus node were confirmed by immunolabelling of two marker proteins, HCN4 and Cx43/Gja1 (Fig. 1b): The pacemaker channel, HCN4, is expressed in the sinus node

but not in atrial muscle and the gap junction channel, Cx43/Gja1, is preferentially expressed in atrial muscle[3]. Frozen biopsies were homogenised and solubilised, followed by protein extraction and enzymatic digestion[4,5]. Peptide samples were pre-fractionated to reduce sample complexity[6,7] and analysed on a high-resolution Q-Exactive HF quadrupole Orbitrap tandem mass spectrometer (LC-MS/MS)[8]. To evaluate technical reproducibility, duplicate experiments were performed for peptide fractionation and LC-MS/MS measurements. Deep proteome measurements identified 7248 proteins stemming from 7025 protein-coding genes (Fig. 1a). All measured proteins are reported in Supplementary Data 1. Sample intensity distributions showed high similarity, with Pearson correlation coefficients >0.9 for all samples (Supplementary Fig. 1a, b). The samples showed high overlap of protein identifications: most proteins were identified in both the sinus node and atrial muscle (>96% overlap in both technical replicates) as well as in all biological replicates (>93% for technical replicate A; >88% overlap for technical replicate B; Supplementary Fig. 1c, d). In a previous study[9], a different strategy was applied to isolate the sinus node, which was subsequently subjected to transcriptomics analysis. Both isolation strategies achieve similar results (Supplementary Fig. 2). We find the transcriptomics and proteomics data to be medium correlated (Supplementary Fig. 3), which is expected, and supports the notion of complementarity of the two sets of information[10,11]. Principal component analysis (PCA) for each technical replicate in our proteomics dataset showed clear distinction of samples into two groups, which were subsequently confirmed to be sinus node and atrial muscle, respectively (Fig. 1c). The majority of variation between samples clearly distinguished sinus node from atrial muscle, explaining 57% and 49% of overall sample variation in the respective technical replicates (component 1, Fig. 1c). As a complementary approach, unsupervised hierarchical clustering also resulted in grouping of sinus node and atrial muscle into two distinct clusters (Fig. 1d and Supplementary Fig. 1e). Given the high-resolution quality and reproducibility of the data across samples, the data forms the basis for quantitative analyses of differences between sinus node and atrial tissue.

**Membrane clock proteins are differentially expressed**. Using intensity-based absolute quantification (iBAQ), we evaluated protein abundances for all proteins involved in the membrane and $Ca^{2+}$ clocks. These measurements compensate for differences in protein size thereby enabling abundance comparisons across proteins[12]. The $Ca^{2+}$ clock proteins were all highly abundant in the sinus node (Fig. 2a, yellow data points), with the central components of the $Ca^{2+}$ clock, Sarco/endoplasmic reticulum $Ca^{2+}$-ATPase (Serca2/Atp2a2), calmodulin (Calm1), phospholamban (Pln), phospholemman (Plm), $Na^+$-$Ca^{2+}$ exchanger (Ncx1/Slc8a1) and ryanodine receptor (Ryr2), being among the top 16% most abundant proteins in the sinus node. All of these proteins were of greater abundance than any membrane clock protein (Fig. 2a, green data points). Of the plasma membrane ion channels involved in the membrane clock, the HCN channels and the $K^+$ inward rectifier ($K_{ir}$) channels were the most abundant in the sinus node. Next, we evaluated differences in protein abundances across the two tissues by analysing label-free quantified mass spectrometry intensities (LFQ)[13]. For each protein, triplicate measurements in sinus node tissue and right atrial tissue were used to calculate a relative fold-change and associated $t$-test-based $p$-value. As illustrated in Fig. 2b none of the $Ca^{2+}$ clock proteins were differentially expressed between the tissues (proteins shown in white). Thus, although the $Ca^{2+}$ clock proteins were highly abundant in the sinus node they were expressed at similar levels in the atrium, which likely reflects the importance of intracellular

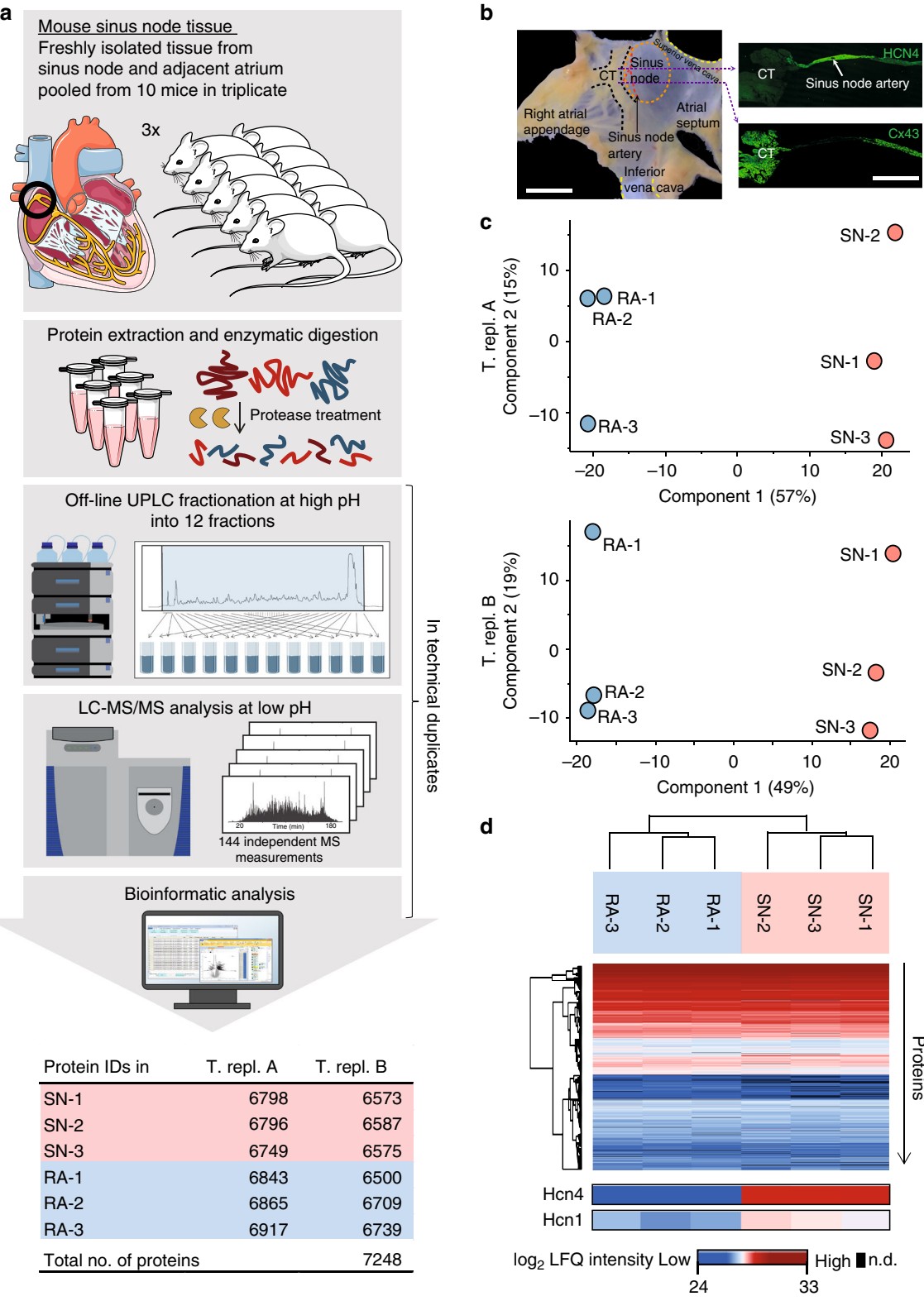

$Ca^{2+}$ handling for the heart as a whole. The membrane clock proteins on the other hand were differentially expressed (Fig. 2b). The HCN4 and HCN1 channels were the most differentially expressed channels in favour of the sinus node (proteins shown in red in Fig. 2b). In addition, the T-type $Ca^{2+}$ channel, $Ca_v3.2$, as well as the two $Ca^{2+}$ channel accessory subunits, $Ca_v\alpha_2\delta_1$/Cacna2d1 and $Ca_v\alpha_2\delta_2$/Cacna2d2, showed higher expression levels in

the sinus node. On the contrary, the inward rectifier $K^+$ channel, $K_{ir}3.1$/Kcnj3, and the two-pore $K^+$ channel, TASK-1/$K_{2P}3.1$, were more highly expressed in the atrial muscle (proteins shown in blue in Fig. 2b). Pacemaking is the result of specialisation of the sinus node. Therefore, the significant differences observed between the sinus node and atrial muscle in membrane clock but not $Ca^{2+}$ clock proteins suggest that it is the unique profile of the

**Fig. 1** High-resolution proteomics measurements of mouse sinus node. **a** Workflow of proteome measurements. Sinus node (SN) and abutting atrial muscle (RA) was collected from ten mice in triplicates. Tissues were homogenised, followed by protein extraction and digestion. The generated peptides were pre-fractionated by UPLC followed by high-resolution LC-MS/MS analysis and bioinformatic data processing. All steps from UPLC fractionation and onwards were performed in technical replicate experiments, T. repl. A and T. repl. B. The experiments resulted in identification of 7248 proteins. **b** Left: Mouse right atrial preparation with intact sinus node showing the crista terminalis (CT), sinus node artery, superior and inferior vena cava, atrial septum and the right atrial appendage. Sinus node tissue biopsies were collected from the area demarcated by the dashed orange line and right atrial samples were collected from the atrial appendage. Right: Atrial tissue sections through the sinus node immunolabelled for hyperpolarisation-activated cyclic nucleotide-gated channel 4 (HCN4, top) and connexin 43 (Cx43, bottom) proteins. Note the small size of the sinus node. Scale bars denote 1 mm. **c** Principal component analysis (PCA) of proteomics data for technical replicates A and B shows that 49–57% of the variation in the dataset is explained by differences in protein expression between sinus node (red) and atrial muscle (blue). **d** Unsupervised hierarchical clustering of protein intensities shows grouping of sinus node and atrial muscle into distinct clusters. Data for HCN proteins is highlighted at bottom. On the colour scale, blue and red indicate low and high protein expression level, respectively. UPLC: ultra-high pressure liquid chromatography; MS: mass spectrometry; LC: liquid chromatography. Source data is provided in Supplementary Data 1

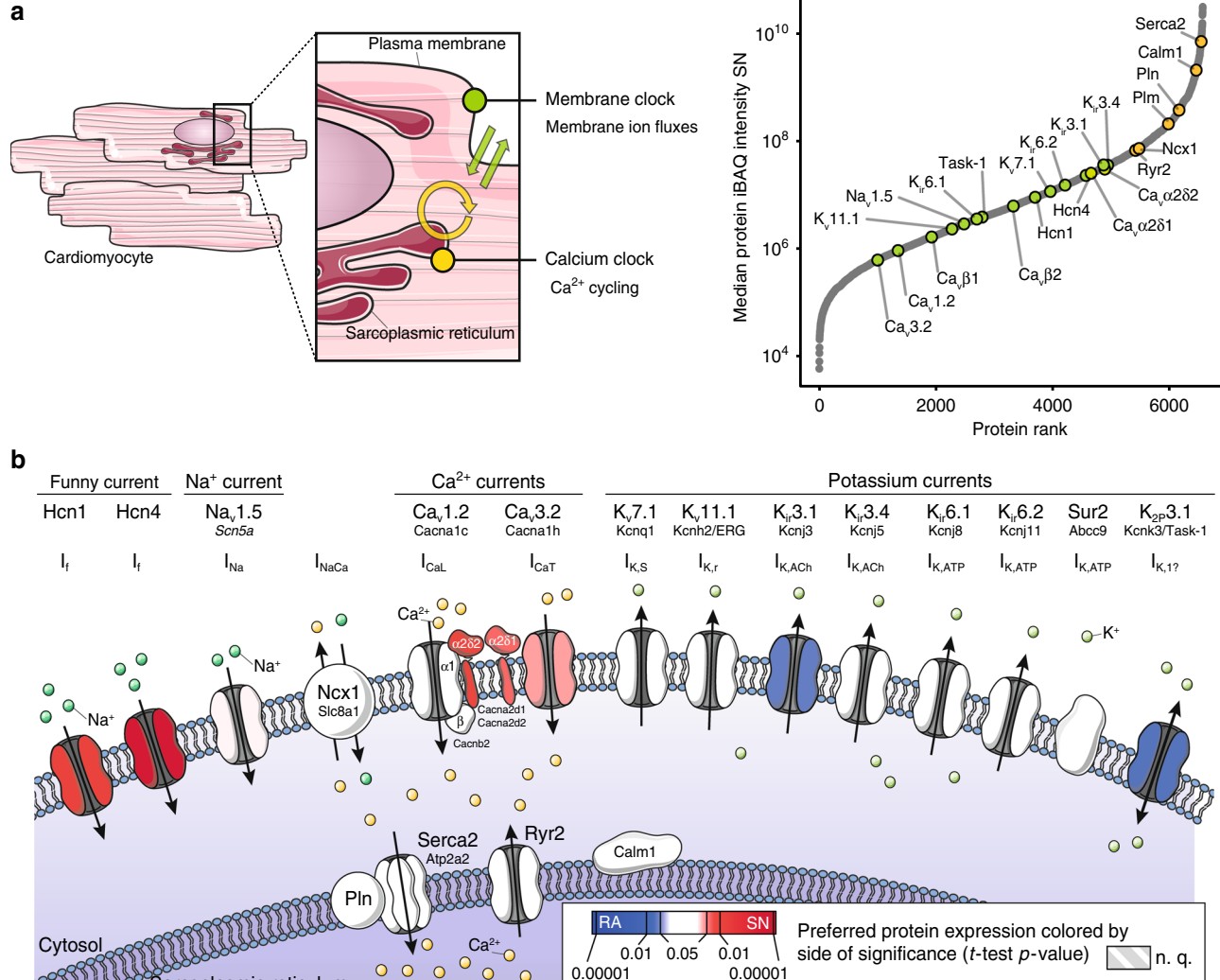

**Fig. 2** $Ca^{2+}$ clock and membrane clock protein expression in the sinus node. **a** Left: Schematic representation of mechanisms contributing to pacemaker action potential generation in the sinus node: membrane ion fluxes referred to as membrane clock are shown in green and $Ca^{2+}$ cycling between sarcoplasmic reticulum, cytoplasm and extracellular space coined $Ca^{2+}$ clock are shown in yellow. Right: Rank plot showing proteins detected in sinus node (x-axis) ranked from lowest to highest mean protein abundance (y-axis). Proteins involved in membrane clock (green) and $Ca^{2+}$ clock (yellow) are highlighted, showing high abundance of $Ca^{2+}$ clock proteins. Soucre Data is provided in Supplementary Data 2. **b** Schematic representation of proteins involved in pacemaker generation in the sinus node. Proteins are coloured by differential expression between sinus node and adjacent atrial muscle according to significance of tissue-specificity (Student's t-test p-value). Red colour represents significantly higher expression in sinus node, blue colour represents significantly higher expression in atrial muscle. Mean values and t-test statistics were obtained from three biological replicates in each group, which each contained pooled tissue sections from ten mice each. N. q.: not quantified; iBAQ: intensity-based absolute quantification. Soucre Data is provided in Supplementary Data 4

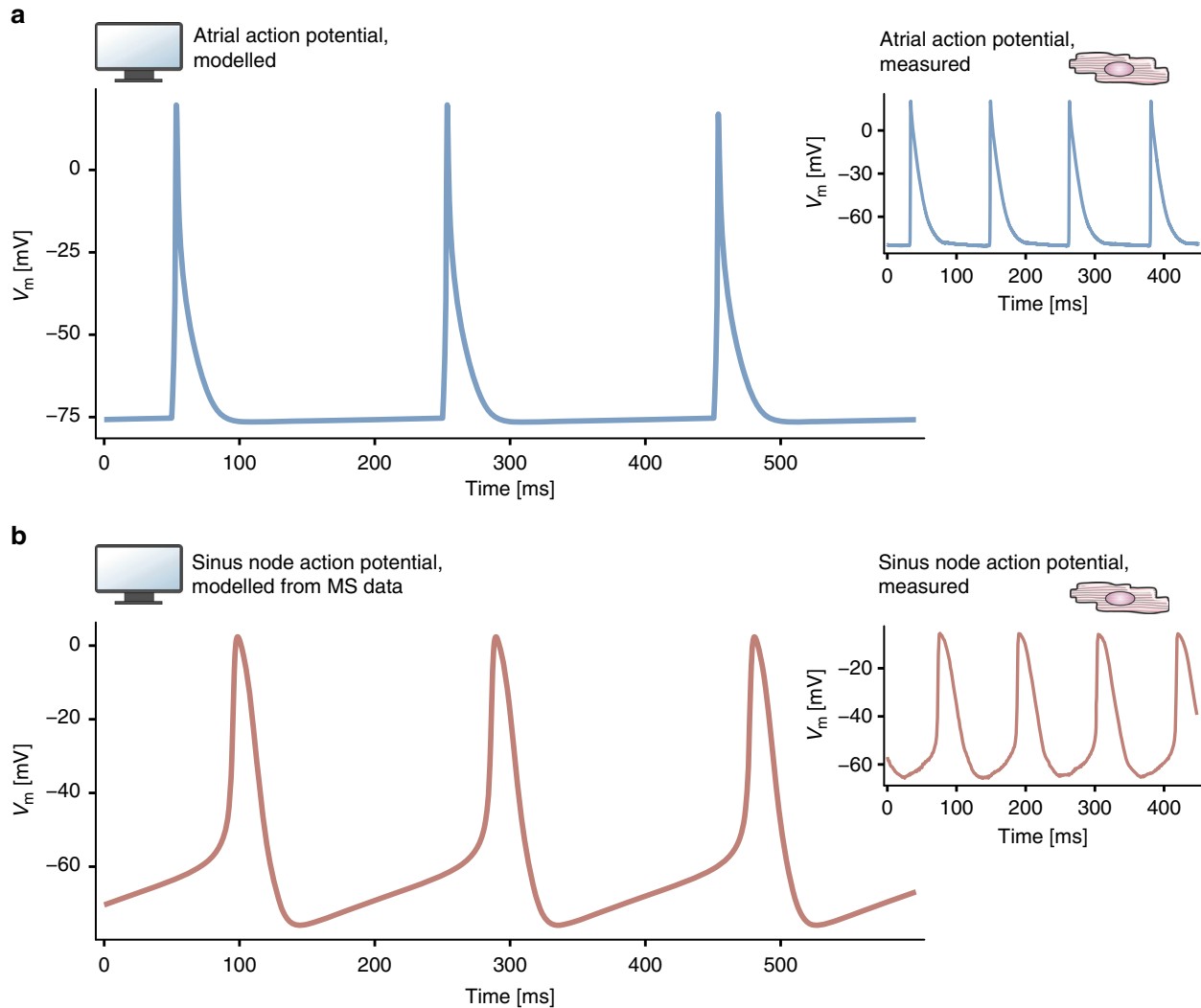

**Fig. 3** Recorded and modelled action potentials from atrial and sinus node myocytes. **a** Atrial action potentials generated by a biophysically detailed model of a mouse atrial myocyte. Inset shows action potentials recorded from mouse atrial muscle with a microelectrode. **b** Action potentials generated by the model when ionic currents were adjusted according to the relative abundance of the corresponding ion channels in the sinus node. The action potential is transformed into a sinus node-like action potential. Inset shows action potentials recorded from mouse sinus node with a microelectrode. MS: mass spectrometry; $V_m$: membrane voltage

ion channels comprising the membrane clock that endows pacemaking.

**Measured abundances recapitulate action potential shape.** Next, computational modelling was employed to test whether the abundance differences in membrane clock proteins between the sinus node and atrial muscle could explain why only the sinus node shows pacemaking. As a reference point, we recorded action potentials from mouse atrial muscle and sinus node using sharp microelectrodes (small inserts in Fig. 3a, b). The atrial action potentials were triggered by stimuli as healthy atrial muscle does not show spontaneous activity, whereas the sinus node pacemaker action potentials were spontaneous. These recordings were subsequently compared to computationally generated action potentials: Fig. 3a shows action potentials generated by a biophysically detailed model of a mouse atrial myocyte developed by Shen et al.[14] in which the action potentials were triggered by stimuli. The model of Shen et al. is the only available model of the mouse atrial myocyte. The atrial action potentials generated by the model are similar to the experimentally recorded atrial action potentials (Fig. 3a). To evaluate if our proteomics-based protein

quantifications can explain the differences between atrial and sinus node action potentials, we scaled all ionic currents involved in the membrane clock in the atrial model according to the protein abundance ratios we had measured by label-free quantitation (Methods and Supplementary Data 1). This resulted in spontaneous sinus node-like action potentials with a prominent pacemaker potential, which resembled the experimentally recorded traces (Fig. 3b). We also did the reverse. In an analogous manner we modified a biophysically detailed model of a mouse sinus node cell developed by Kharache et al.[15] We were able to successfully convert the sinus node action potential to an atrial-like action potential (non-spontaneous, triangular action potential shape; see Methods and Supplementary Discussion for further details). We conclude that the differences in expression of ion channels comprising the membrane clock can explain why the sinus node shows pacemaking.

**Quantification of more than 50 ion channel subunits.** In total, our deep proteome measurements detected >50 ion channel subunits in the sinus node and atrial muscle (Fig. 4 and Supplementary Data 2). This greatly expands the number of ion

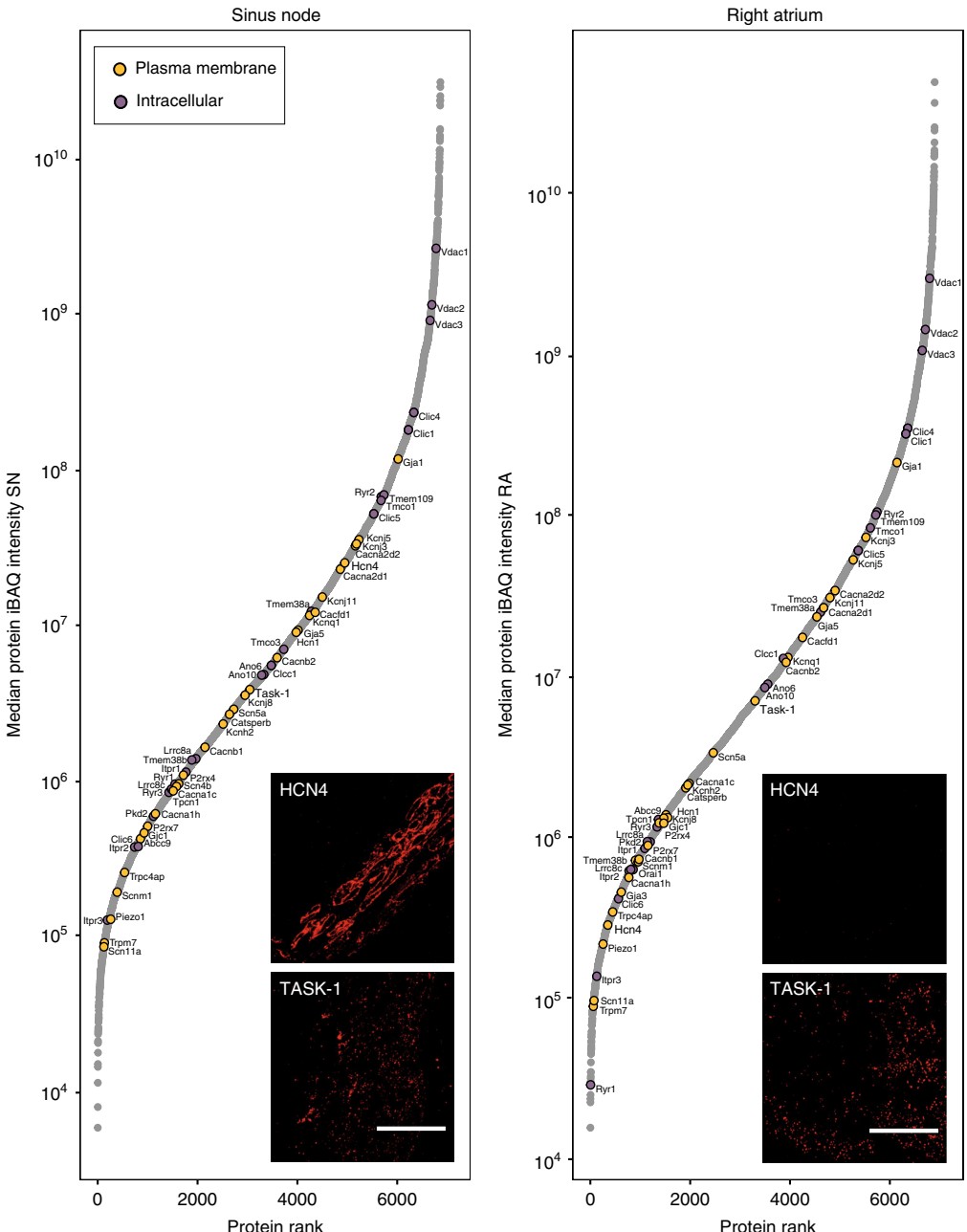

**Fig. 4** Abundance of ion channels in sinus node and atrial muscle. Rank plots show proteins ranked from lowest to highest protein intensity (*y*-axis) for sinus node (SN, left) and atrial muscle (RA, right). Ion channels of interest are highlighted and classified by major subcellular localisation (plasma membrane or intracellular). Image insets show immunolabeling of Hcn4 or TASK-1 in tissue sections from sinus node (left) or atrial muscle (right), respectively. iBAQ, intensity-based absolute quantification. Scale bars denote 50 μm. Mean protein intensity values are based on three biological replicates in each group, each containing pooled tissue sections from ten mice. Source Data is provided in Supplementary Data 3

channel subunits known to be expressed in these tissues. We identified ion channel subunits among the lowest abundant proteins in the heart tissues all the way to the most abundant proteins (Fig. 4). Plasma membrane ion channel subunits identified include well-known voltage-gated ion channel subunits: funny channels (HCN1, HCN4), $Na^+$ channels ($Na_v$1.5/Scn5a, $Na_v$1.9/Scn11a), $Ca^{2+}$ channels ($Ca_v$1.2/Cacna1c, $Ca_v$3.2/Cacna1h, $Ca_v\alpha_2\delta_1$/Cacna2d1, $Ca_v\alpha_2\delta_2$/Cacna2d2, $Ca_v\beta_{1-3}$/Cacnb1-3), delayed rectifier $K^+$ channels ($K_v$11.1/Kcnh2, $K_v$7.1/Kcnq1) and inward rectifier $K^+$ channels ($K_{ir}$3.1/Kcnj3, $K_{ir}$3.4/Kcnj5, $K_{ir}$6.1/Kcnj8, $K_{ir}$6.2/Kcnj11). Of intracellular ion channel subunits, we identified ryanodine receptors (Ryr1-3) and inositol 1,4,5-trisphosphate receptors (Itpr1-3) responsible for SR $Ca^{2+}$ release

and, in addition, potential SR ion channel subunits responsible for maintaining electroneutrality on uptake and release of $Ca^{2+}$ into and from the SR (TricA/Tmem38a, TricB/Tmem38b, Mg23/Tmem109 and Clic[16]). Also mitochondrial ion channel subunits (mitochondrial $Ca^{2+}$ uniporter/MCU, Vdac and Clic[17]) were detected. We further detected less studied cardiac ion channel subunits: Orai1, Stim1, P2xr4, P2xr7, TASK-1/Kcnk3 and Trpm7. Orai1 and Stim1 have been shown to be responsible for a store-operated $Ca^{2+}$ current in the sinus node[18]. We have previously shown expression of the purinergic receptor ligand gated ion channels, P2xr4 and P2xr7, in rat and human sinus node[19]. TASK-1/Kcnk3 (twin pore $K^+$ channel) loss-of-function mutations promote atrial fibrillation[20]. TASK-1/Kcnk3 has not

previously been shown to be expressed in sinus node; in Fig. 4, immunolabelling is shown for TASK-1 in the mouse sinus node. Trpm7 has previously been shown to be essential for cardiac pacemaking[21] and to be responsible for fibrosis in sick sinus syndrome[22].

To drive the atrial muscle, the sinus node is electrically coupled to the atrial muscle via gap junction channels. The atrial muscle electrically suppresses the pacemaker activity of the sinus node[23], and theoretical studies have shown that electrical coupling in the sinus node must be poorer than in atrial muscle for it to drive the atrial muscle but not be suppressed by it[24,25]. This is indeed what our data shows: the medium and high conductance connexins (Cx43/Gja1 60–100 pS[3] and Cx40/Gja5 200 pS[3]) were the most abundant connexins, and were less abundant in the sinus node than in the atrial muscle (Supplementary Fig. 4). The low conductance Cx45/Gjc1 (20–40 pS[3]) was less abundant and uniformly expressed, and the high conductance Cx46/Gja3 (140 pS[26]) was poorly and uniformly expressed. These data are consistent with poorer electrical coupling in the sinus node than in the atrial muscle.

Taken together, this dataset presents the most comprehensive quantitation of cardiac ion channel protein expression in the sinus node.

**Ion channel copy numbers in sinus node myocytes**. Based on the unique protein expression data for ion channel subunits combined with the observation that their abundance varied enormously, we set out to determine copy numbers of ion channels per myocyte in the sinus node. Historically, measurements of ion channel number per myocyte have been difficult to obtain, and to our knowledge only two estimates of such numbers exist: the number of L-type $Ca^{2+}$ channels per ventricular myocyte has been estimated, but the findings are 8-fold different, 27,030 channels per myocyte and 215,380 channels per myocyte[27,28]. Based on our measurements, the estimated channel copy numbers will rely on the assumption that the ion channel measurements originate from myocytes and the majority of the subunits reside in the plasma membrane. Is it reasonable to assume that ion channel abundances originate from myocytes? Currently, we cannot measure deep proteomes of isolated sinus node myocytes to evaluate this hypothesis due to the limited number of cells. We reasoned that the only other cardiac cell type that could substantially contribute to channel abundance would be fibroblasts, due to their high abundance in the heart. Thus, to query for ion channel protein abundance in cardiac fibroblasts we isolated primary cardiac fibroblasts and measured a proteome at a depth of >5500 proteins (details provided in the Methods). Only 20 channels were measured in fibroblasts (Supplementary Fig. 5). Importantly, none of the channels of focus here were among these 20 channels. Hence, we consider the assumption that ion channel abundances originate from myocytes to be reasonable. We evaluated the relative abundance of all ion channel subunits from iBAQ[12]. For each ion channel, we calculated the corresponding abundance by adjusting for the number of subunits constituting a functional channel, such as four for HCN4 and one for $Ca_v1.2$/Cacna1c (Supplementary Data 3). The estimated copy numbers for selected ion channels are shown in Fig. 5a. To determine the absolute number of ion channels per myocyte, data were normalised to an independent estimate of the number of HCN4 channels per myocyte, which we obtained from a Markov chain model of HCN4 current shown in Fig. 5b–e. The Markov model was developed based on the best model available of the mouse sinus node cell action potential[15], where the HCN4 channel copy number was estimated considering open and closed states, stochastic behaviour and single channel conductance (see

Supplementary Fig. 6). This provided an estimate of 6,255 HCN4 channels per sinus node myocyte. We used this number to convert the relative ion channel abundances based on mass spectrometry measurements into an absolute number of channels per myocyte (Fig. 5a, extended information in Supplementary Fig. 7, and Supplementary Data 3). Our calculations suggest the presence of, for example, an average of 2227 HCN1 channels and 9438 $K_{ir}3.4$/Kcnj5 channels in each sinus node myocyte. The number of plasma membrane ion channels per myocyte is dwarfed by the number of intracellular ion channels—for example, we find 16,079 Ryr2 SR $Ca^{2+}$ release channels and 6,630,737 SR Serca2 $Ca^{2+}$-ATPases per myocyte. For comparison, we also estimated ion channel copy numbers for HCN1, $Ca_v1.2$, $Ca_v3.2$ and $K_v11.1$ from Markov chain models (Supplementary Figs. 8–11), which resulted in qualitatively similar numbers as derived by our mass-spectrometry-based approach (Supplementary Fig. 12). The stoichiometric relationship between plasma membrane and intracellular Ca2+-handling proteins is considered further in the Supplementary Discussion. This is a first estimate of copy numbers of ion channel subunits, exchangers and pumps per myocyte in the sinus node.

**Functional differences of nodal and atrial protein profiles**. The proteomics data covers the entire protein landscape of the sinus node, and thereby offers the possibility to ask which proteins are differentially expressed between sinus node and atrial muscle for the whole dataset, and what biological pathways these proteins represent. For all measured proteins, we analysed for differential expression between the two tissues based on LFQ intensities. Volcano plot analysis of the entire dataset revealed 575 proteins with statistically significant differential expression between the sinus node and atrial muscle (Fig. 6a and Supplementary Data 4). In Fig. 6a, proteins more highly expressed in the sinus node are shown in red and those more highly expressed in the atrial muscle are shown in blue. This analysis confirms the well-known prominent protein expression profiles of known tissue markers such as the pacemaker channels, HCN1 and HCN4, for the sinus node, and the gap junction channels, Cx40/Gja5 and Cx43/Gja1, and the natriuretic peptides, Nppa (atrial natriuretic peptide) and Nppb (brain natriuretic peptide), for the atrial muscle (Fig. 6a, b). Notably, the pacemaker channel HCN4 was the most differentially expressed protein between tissues. In addition to confirming previous findings on sinus node protein expression, our data highlights >500 proteins that were previously unrecognised to be differentially expressed between the sinus node and atrial muscle.

To identify biological pathways represented by the 575 proteins that distinguish sinus node tissue from atrial tissue, we generated a protein interaction network of all significantly differentially expressed proteins in the sinus node and atrial muscle (Fig. 7 and Supplementary Data 5). In general, proteins more highly expressed in the sinus node preferentially interacted with other sinus node-specific proteins, and similarly proteins more highly expressed in the atrial tissue formed clusters with other atrial-specific proteins. Proteins with higher expression in atrial muscle (shown in blue in Fig. 7) formed clusters enriched for exocytosis, translation, cell adhesion, integrin signalling and cell activation. The general over-representation of various components of the natriuretic peptide system and proteins associated with exocytosis are further presented in Supplementary Fig. 13. Proteins with higher expression in the sinus node (shown in red in Fig. 7) formed clusters enriched for actin cytoskeleton, contractile fibres, chromatin, neurofilament, carbohydrate metabolism, lipid metabolism, collagen and connective tissue. Taken together, these clusters represent the unique sinus node architecture at the molecular level.

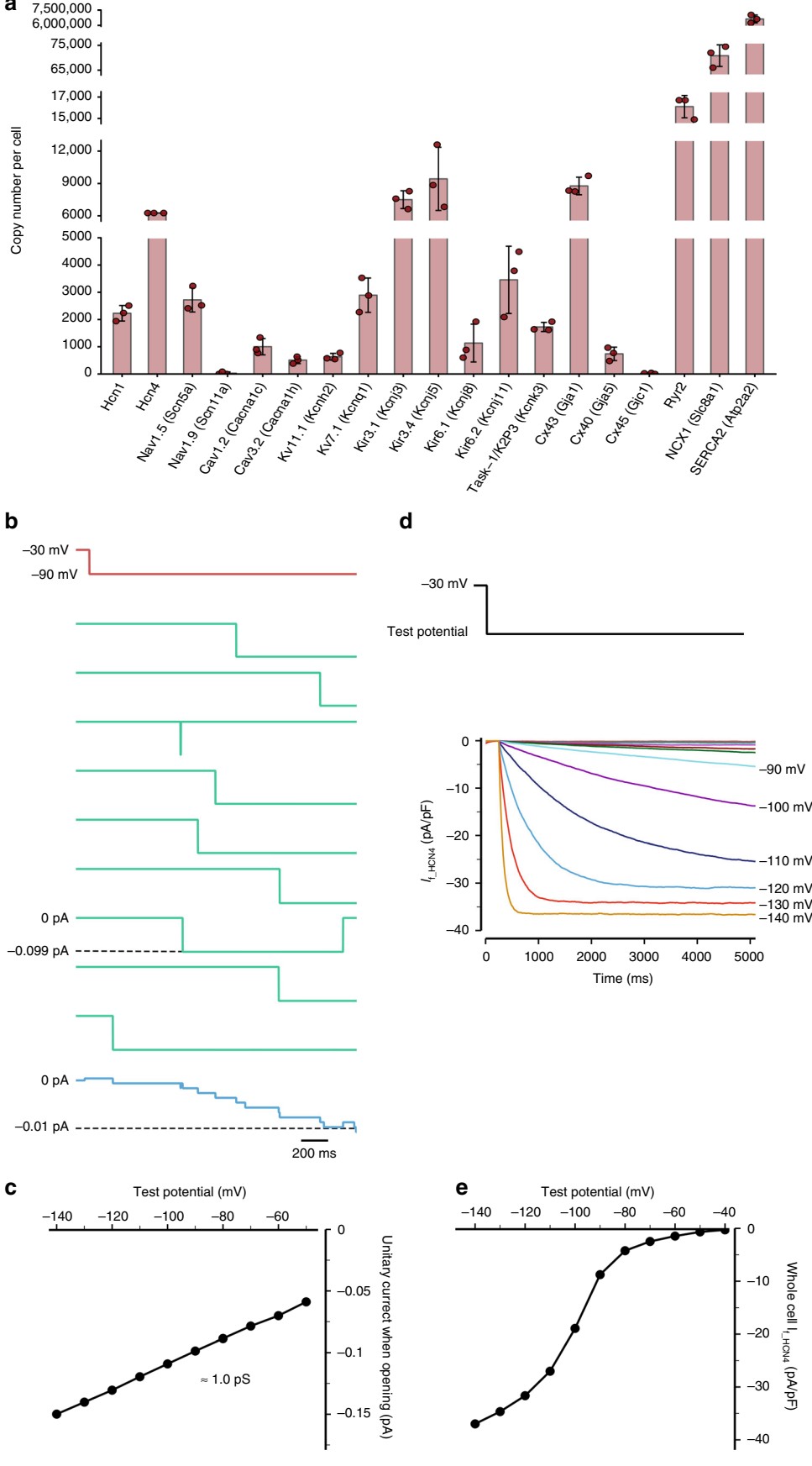

**Fig. 5** Estimation of ion channel numbers in a sinus node cell. **a** Estimated number of ion channels in a sinus node cell based on mass spectrometry. Calculation of ion channel numbers is based on measured relative ion channel protein abundance taking numbers of subunits into consideration; absolute ion channel number was then calculated by normalising to the number of HCN4 channels estimated at 6255 HCN4 channels per nodal cell from the Markov chain model below. Error bars indicate mean +/− standard deviation. Data points indicate underlying single values. Mean protein intensity values are based on three biological replicates in each group, each containing pooled tissue sections from ten mice. Source data are provided as a Source Data file. **b**–**e** Markov chain model of HCN4 current. **b** Unitary current traces (green; 9th, 13th, 24th, 25th, 38th, 54th, 69th, 73th and 79th sweeps) and ensemble average current of 100 consecutive sweeps (blue) during a hyperpolarising pulse from a holding potential of −30 mV to −90 mV for 2000 ms (red). **c** Current–voltage relationship of the unitary HCN4 channel. Solid circles show the maximum current when the channel was open during ten sweeps. The slope of the line shows the unitary channel conductance to be ~1.0 pS. **d** Whole cell current from 6255 unitary HCN4 channels with a holding potential of −30 mV and test potentials from −140 mV to −40 mV. **e** Whole cell HCN4 current–voltage relationship of the whole cell HCN4 current. The maximum current was recorded at 10 s after the application of the test pulse

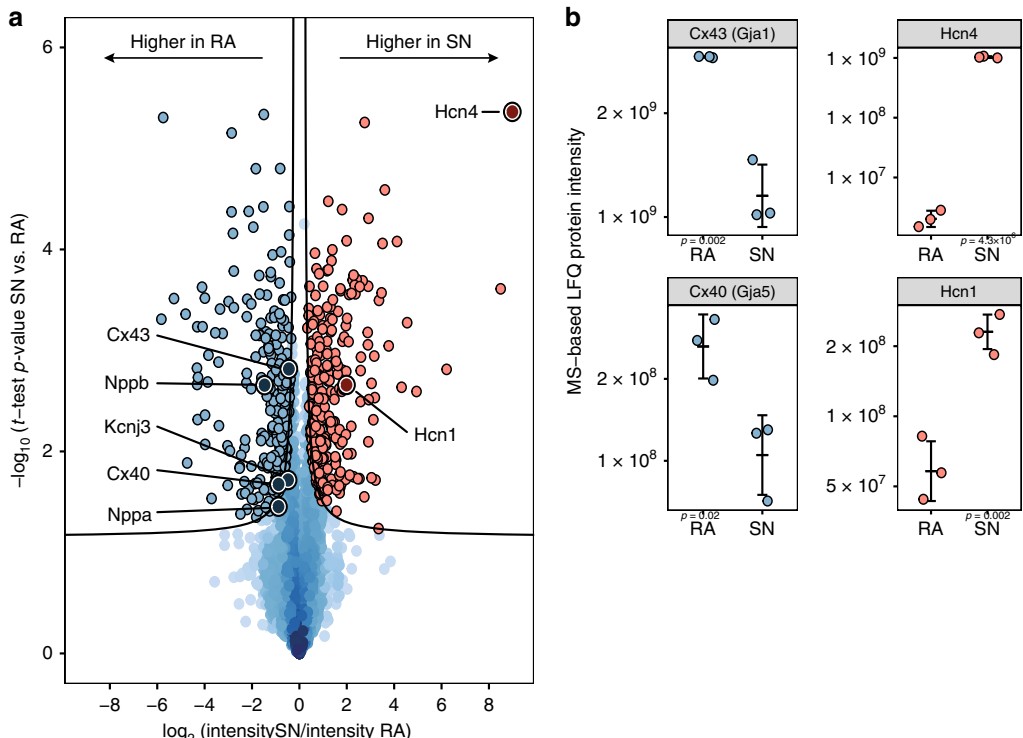

**Fig. 6** Differentially abundant proteins between sinus node (SN) and atrial muscle (RA). **a** Volcano plot analysis revealed 575 significantly differentially expressed proteins at 5% false-discovery rate (FDR). Two-hundred ninety-one proteins were of higher abundance in the sinus node (red points) and 284 of higher abundance in the atrial muscle (blue points). The FDR cutoff is indicated by black lines. Prominent marker proteins for each tissue are highlighted. Source Data is provided in Supplementary Data 4. **b** Jitter plot representation of selected proteins exemplifying label-free quantitation (LFQ) measurements underlying the volcano plot analysis. Examples of the individual mass spectrometry-based intensity measurements behind each point in the global analysis represented in **a** are shown for HCN1, HCN4, Cx40/Gja5 and Cx43/Gja1. *T*-test-based *p*-values are indicated in the plots. Error bars indicate mean +/− standard deviation. Data points indicate underlying intensity values. Source data are provided as a Source Data file. Mean protein intensity values are based on three biological replicates in each group, each containing pooled tissue sections from ten mice

**Electron micrographs visualise cellular composition**. To better interpret the protein network findings, we used electron microscopy to evaluate the structural characteristics of the sinus node and atrial muscle (Fig. 8 and Supplementary Figs. 14–16; Supplementary Movies 1 and 2). Contractile proteins were enriched in the sinus node (Fig. 7) and yet classically sinus node cells are regarded as empty, containing relatively few myofibrils[29]. However, consistent with the proteomics data, electron micrographs showed that sinus node cells contain abundant myofilaments; yet these are poorly organised and not associated into myofibrils (Supplementary Fig. 14a–d). In contrast, atrial myocytes have well organised myofibrils with bundles of alternating myosin and actin filaments and prominent Z-lines generating a striated pattern along the length of the myocyte (Supplementary Fig. 15).

The over-representation of metabolic proteins in the sinus node indicates metabolic specialisation of the tissue. Enrichment analyses showed that proteins with higher sinus node expression were mainly associated with anabolic processes, including lipogenesis, gluconeogenesis and ketogenesis. In particular, proteins involved in lipid synthesis (FASN, Acly, Thrsp, Acaca, Retn and Dgat1) and lipid storage (Plin1) had the greatest fold-difference in expression (Supplementary Fig. 17). Most of these proteins are exclusively or predominantly expressed in adipose tissue[30]. Thus, lipid storage may be more common in the sinus node than in atrial muscle. This is significant because fatty acids are the major source of energy for the heart and this may be related to the sinus node being the ultimum moriens according to Wenckebach[31]. Consistent with this, electron micrographs showed perinuclear lipid vesicles in sinus node myocytes (and atrial myocyte; Supplementary Figs. 14c and 15g), as well as adipocytes in the sinus node. The sinus node is characterised by a haphazard array of small (5–10 μm) spindle-shaped sinus node

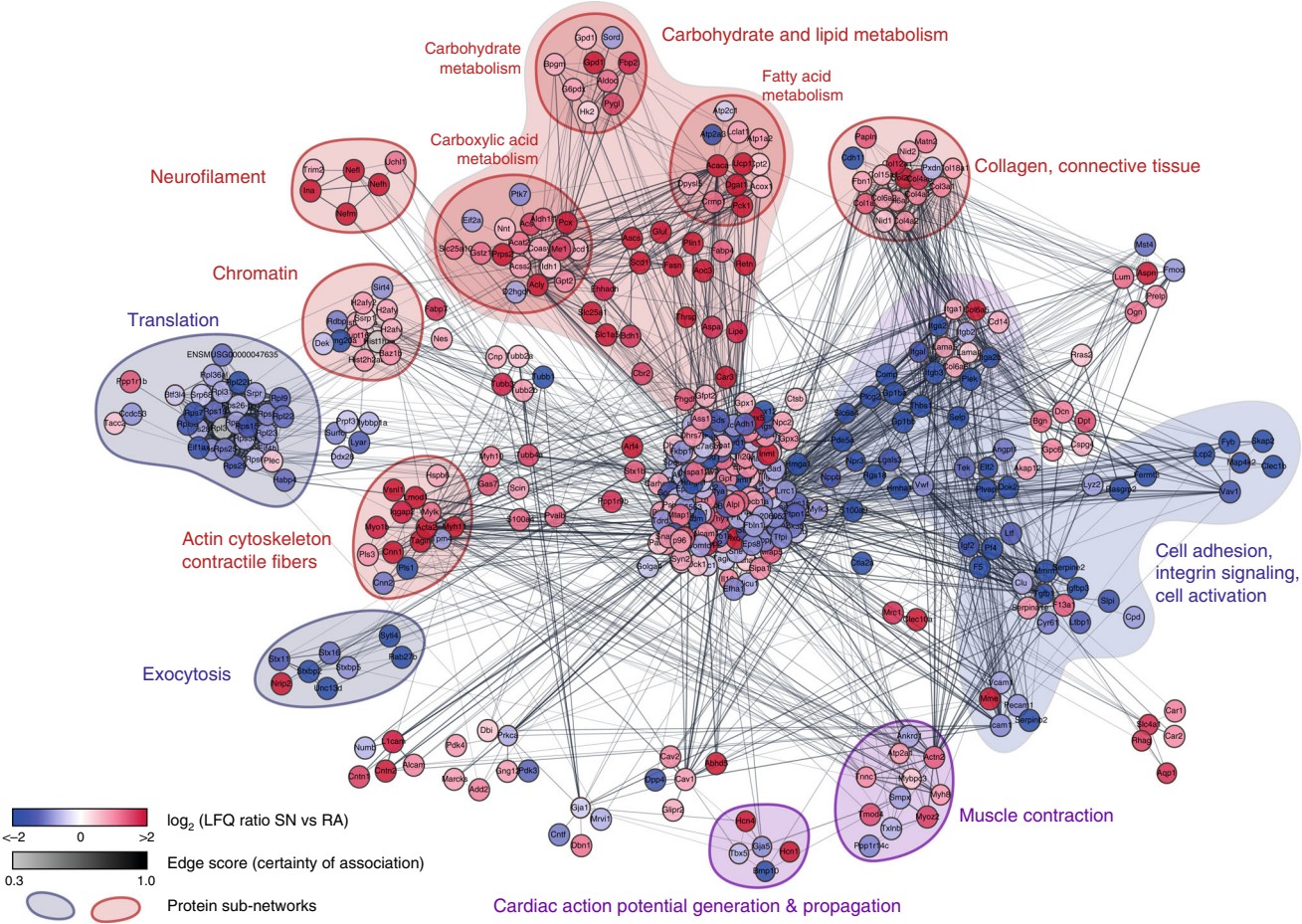

**Fig. 7** Protein network analysis of significant proteins in sinus node (SN) versus atrial muscle (RA). A STRING network was retrieved for all proteins deemed significant by volcano plot analysis. Clustering was performed relative to confidence score of the interaction, and ontology enrichment analysis was performed on the resulting clusters. Gene Ontology terms describing the clusters were deduced as highlighted in the figure, pointing to overall differences between sinus node and atrial muscle. Colours represent tissue of overexpression (red—SN, blue—RA) and cluster colours represent the tissue-specificity of the cluster. Mean protein intensity values are based on three biological replicates in each group, each containing pooled tissue sections from ten mice

myocytes, but other cell types are also present (Fig. 8a) and they leave their signature in the proteomics data. Neurofilament proteins (forming intermediate filaments in autonomic axons[32]) were more highly expressed in the sinus node (Fig. 7). Consistent with this, electron micrographs revealed a high abundance of autonomic nerve fibres in the sinus node and few in the atrial muscle (Fig. 8a, b and Supplementary Figs. 14a, e and 15c). Electron micrographs showed that in the sinus node (and atrial muscle) the epicardium is lined with numerous finger-like cilia (Fig. 8a, b and Supplementary Fig. 15a, b). Kinesin II (Kif3)[33] and tubulin (Tuba)[34] are essential constituents of cilia, and the presence of cilia is reflected in measurements of Kif3a, Kif3b, Kif3c, Tuba1a, Tuba1b, Tuba1c, Tuba3a, Tuba4a and Tuba8 in the sinus node proteome (Supplementary Data 1). Macrophages have been reported in the atrioventricular node and shown to be functionally important for conduction[35] and macrophages are evident in the electron micrographs of the sinus node (Fig. 8a and Supplementary Fig. 14g). Consistent with this, CD163, a marker of M2 macrophages[35,36] was present in the sinus node (Supplementary Data 1). In addition, the sinus node artery (with associated cell types), telocytes (novel type of interstitial cell reported in the human sinus node[37]), monocytes and fibroblasts were observed in the electron micrographs of the sinus node (Supplementary Fig. 14g). The fibroblasts help in the formation of the extracellular matrix and the electron micrographs showed

that the sinus node cells (unlike atrial myocytes) are loosely packed within copious extracellular matrix, including basement membrane, elastin and wavy bundles of collagen fibres (Fig. 8a and Supplementary Fig. 16).

**Extracellular matrix with highly elastic properties**. Both the sinus node and atrial muscle samples were enriched in extracellular matrix (ECM) glycoproteins and proteoglycans (see detailed analysis in Supplementary Fig. 18). In contrast to typical mesenchymal ECM, the most abundant components were basement membrane and elastic microfibril macromolecules. On the electron micrograph, basement membrane is seen encompassing individual cardiomyocyte (Supplementary Figs. 14d and 15g). All major components of basement membranes were detected in the proteomics assessment, including the network-forming laminins and collagen type IV, together with nidogens-1 and −2 and the heparan sulfate proteoglycan perlecan. Fibronectin, which is an abundant ECM glycoprotein peripherally associated with basement membranes, was also enriched in sinus node. Immunohistochemical staining of both sinus node and right atrial tissue with antibodies directed against either laminin or collagen IV revealed thin, sheet-like ECM surrounding individual myocytes typical of basement membrane protein distribution seen on the electron micrograph and those previously reported for skeletal and cardiac

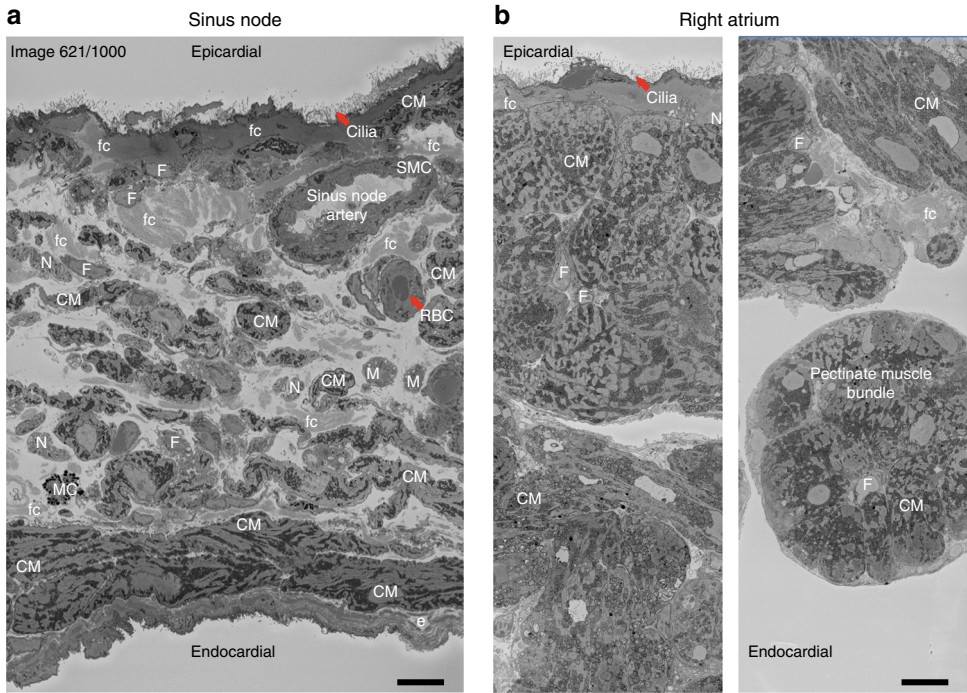

**Fig. 8** Electron micrograph of mouse sinus node and right atrium. **a** Transmission electron micrograph showing cross sectional view of the centre of sinus node with sinus node artery. Cells are labelled in abbreviated capital letters: CM, cardiomyocyte; N, nerve; SMC, smooth muscle cell; RBC, red blood cell; M, macrophage; F, fibroblast; MC, mast cell. ECM components labelled in abbreviated small letters: fc, fibrous collagen; e, elastin. **b** Transmission electron micrographs showing the epicardial (left) and endocardial (right) regions of the right atrial appendage. Scale bars denote 10 μm

muscle[38–40] (Supplementary Fig. 19). The most abundant collagen in sinus node ECM was type VI, followed by types XII, XV and XVIII. Collagen VI is normally a heterotrimer of α1, α2 and α3 subunits, but intriguingly, the α3 subunit was not detected and instead the α6 and, to a lesser extent, α5 subunits were present. The latter subunits have been reported to substitute for α3 in some tissues, including skeletal muscle[41]. Collagen VI forms beaded microfibrils and may function together with fibrillin, TGFβ-binding proteins and elastin to endow sinus node tissue with elastic properties (Supplementary Fig. 16c–f). Unlike collagen VI, most elastic proteins did not show enrichment to sinus node tissue, although tissue staining with antibodies directed against Mfap2 (MAGP1) and elastin revealed a more elaborate staining in sinus node tissue, with the right atrium being more densely packed with cells (Supplementary Fig. 19). On the electron micrograph a layer of elastin lines the sub-endocardial surface of the sinus node (Fig. 8a and Supplementary Fig. 16d–f). Sinus node ECM was characteristically enriched with a large number of small, leucine-rich repeat proteoglycans (SLRPs). These proteoglycans are common constituents of interstitial matrices and basement membranes where they play key roles in fibrillogenesis, TGFβ sequestration and Ca$^{2+}$ binding[42–44]. Finally, consistent with the enrichment for basement membrane, fibronectin and collagen VI in sinus node ECM, integrin subunits α1 (collagen-binding), α7 (laminin-binding) and α9 (fibronectin-binding) were more abundant in the sinus node samples. Taken together, these findings define the unique ECM composition with highly elastic properties in the sinus node.

**snRNA-seq of sinus node links proteins to cell types**. From our quantitative proteomics analysis of the sinus node and atrial tissues we have identified specific protein differences between these tissues. As evidenced by the electron microscopy data presented above, it is challenging to attribute the protein differences to a particular cell type due to the cellular complexity of the

tissue. To overcome this challenge, we performed snRNA-seq of mouse sinus node biopsies. Specifically, sinus node tissue was isolated in a similar manner as described for the proteomics experiments, and for each replicate, sinus nodes from four mice were pooled. We extracted nuclei from the biopsies and prepared single-nuclei suspensions for snRNA-seq analysis. We chose this preparation protocol to minimise transcriptional profile alterations during sample handling. We obtained single-nucleus libraries using the Chromium system (10x Genomics) and sequenced them on a Illumina NovaSeq 6000. After quality control, the dataset comprised 5357 nuclei transcriptomes and covered a total of 27,998 genes, with a per cell average of 2715 expressed genes. The dataset of 27,998 transcripts are provided in Supplementary Data 6. This data enabled us to undertake a comprehensive transcriptomics characterisation of the mouse sinus node. When projecting the nuclei to a two-dimensional t-distributed stochastic neighbour embedding (tSNE) plot, we observed a clear distinction between cell types of the sinus node identifying 12 distinct cell clusters (Fig. 9a). As expected, the clusters comprised sinus node myocytes and fibroblasts. We also identified macrophages, endothelial and endocardial cells and epithelial and epicardial cells. Interestingly, we also found three cell clusters with gene profiles suggesting subpopulations of adipocytes expressing marker genes indicating differing levels of thermogenic activity (Fig. 9b). Our snRNA-seq data illustrates the cellular complexity of the sinus node tissue beyond what could be inferred from the electron micrographs. Importantly, we can utilise the snRNA-seq data in the interpretation of our protein-based findings. For instance, our ion channel copy numbers estimate relied on the assumption that the channels investigated are exclusively expressed in the sinus node myocytes. From the snRNA-seq data we extracted information on which sinus node cell types express the ion channel mRNA and found that the membrane clock proteins were indeed predominantly expressed in the sinus node myocytes (Supplementary Fig. 20). For any

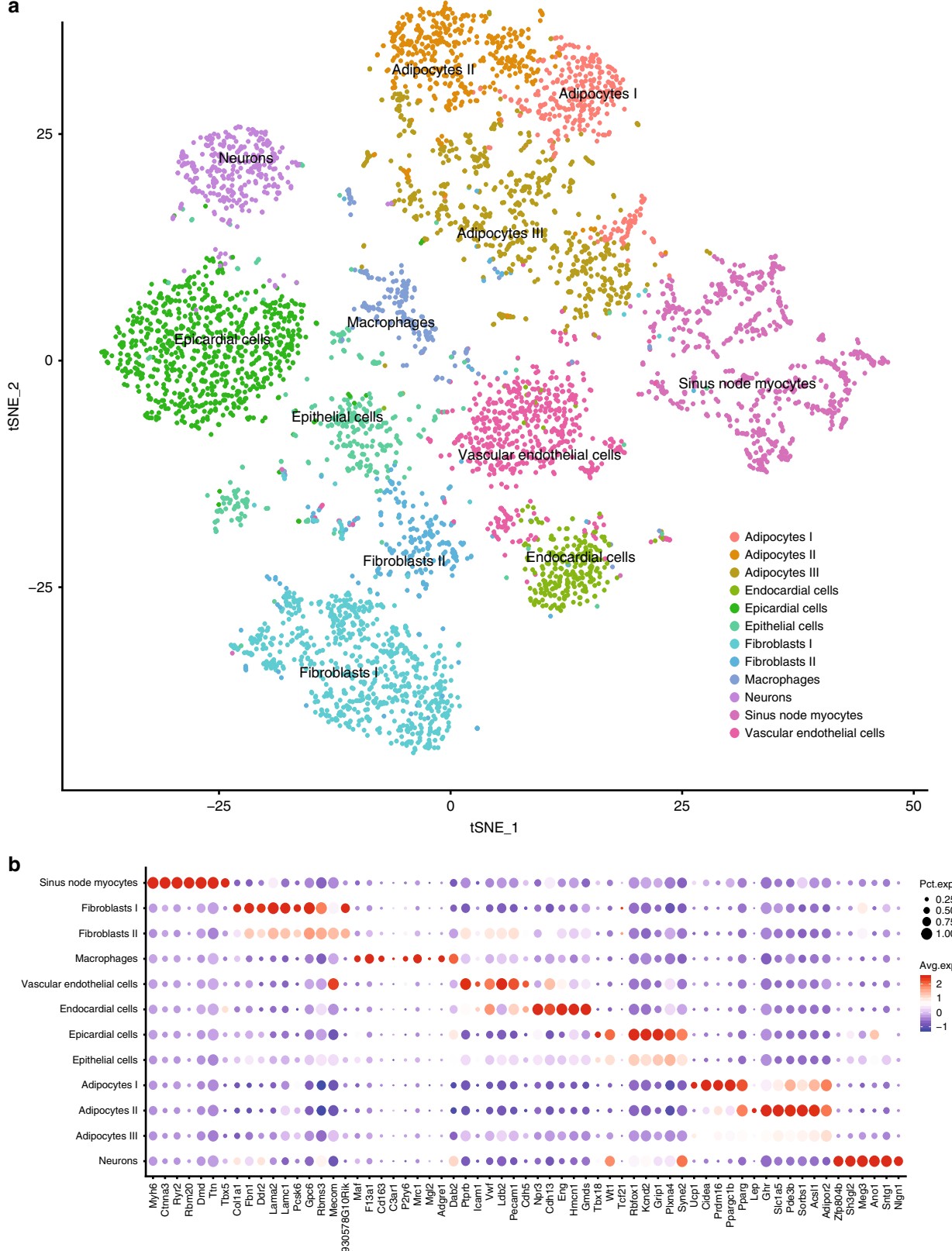

**Fig. 9** Single-nucleus RNA sequencing analysis of sinus node tissue reveals the major cell types. **a** Two-dimensional t-SNE (Stochastic Neighbour Embedding) plot outlines the major cell populations of the sinus node. Each point represents a single nucleus. Cell populations are coloured according to cluster designation. **b** Centred and scaled expression levels of cell cluster markers for sinus node myocytes, fibroblasts, macrophages, vascular endothelial cells, endocardial cells, epicardial cells, adipocytes and neurons. Mean values are based on four biological replicates in each group, each containing pooled tissue sections from four mice. Source Data is provided in Supplementary Data 6 and 7

protein discussed in this paper, the snRNA-seq dataset is provided as a reference point to identify the cell population expressing the protein (Supplementary Datas 6 and 7).

## Discussion

This study provides a comprehensive investigation of the proteome of the natural pacemaker of the heart, the sinus node. Over 7000 proteins were quantified making it one of the most detailed quantitative cardiac proteomes to date. It is the only proteome of the sinus node, which makes it an important addition to other current cardiac protein atlases of the heart[45,46]. The sinus node has previously been investigated at the transcriptome level[9]. We present snRNA-seq analysis of the sinus node, providing a map of its cell type composition. Combining the information from the quantitative proteomics dataset with the information on transcripts across cell types enable us to identify the particular cellular population underlying the protein differences between sinus node and atrial tissues. We showed that, ~8% of all quantified proteins were differentially expressed between the sinus node and atrial muscle. The deep tissue proteome measurements of the sinus node revealed molecular specialisations: For example, in the sinus node: (1) pacemaking ion channels are more highly expressed; (2) gap junctions are more poorly expressed; (3) contractile proteins linked to sinus node dysfunction are more highly expressed; (4) proteins of the natriuretic peptide system are more poorly expressed; (5) proteins involved in lipid storage are more highly expressed; (6) there is a unique expression pattern of transcription factors; (7) there is more neuronal innervation; and (8) the extracellular matrix is characteristic of highly elastic tissue and presumably affords protection to the sinus node, which is located in the thin rear wall of the right atrium and subject to regular distension as blood moves in and out of the chamber. For each of these functional specialisations, the cell type underlying the signal can easily be identified by querying our snRNA-seq dataset with regards to cell populations expressing the corresponding transcript. For instance, the proteomics dataset has yielded the relative abundances of proteins, such as plasma membrane as well as intracellular ion channels, exchangers and pumps. For the ion channels contributing to the membrane clock we show that they are exclusively expressed in the myocytes of the sinus node. This type of quantitative information will be the foundation of more detailed modelling of the heart than is currently possible. We have exemplified this potential by predicting a sinus node-like action potential based on the proteome data. Our dataset contributes to the long-lasting dispute over the roles of the membrane and $Ca^{2+}$ clocks in pacemaking. The $Ca^{2+}$ clock regulates sinus node pacemaker activity through $Ca^{2+}$ cycling in the SR[47,48]. We detected $Ca^{2+}$ clock proteins at high expression levels and uniformly expressed in the sinus node and atrial muscle. Their high abundance is consistent with previous reports on transcript level[49] (which however suggested differential expression of some of the components). Our protein data points to a central but uniform role of the $Ca^{2+}$ clock between pacemaking sinus node and non-pacemaking atrial muscle. Nevertheless, it is possible that differential expression of some related but less studied proteins plays a physiologically important role in $Ca^{2+}$ cycling in the sinus node, such as the regulatory proteins protein kinase A (Pka), $Ca^{2+}$-calmodulin-dependent protein kinase II (Camk2) and phosphodiesterase 1A (Pde1a)[48,50]. However, we found Pka and Camk2 to be more abundant in the atrial muscle, and the phosphodiesterases detected were either uniformly expressed or more highly expressed (Pde3a and Pde5e) in the atrial muscle. The data presented here provides a unique picture of plasma membrane ion channels, which together comprise the membrane clock. Some of the findings confirm prior knowledge,

but many are novel. The higher abundance of the pacemaker channels, Hcn1 and Hcn4, in the sinus node has previously been reported at both the transcript level[9] and the protein level[51] and as such is a validation of the dataset. The higher expression of Hcn1 and Hcn4 in the sinus node is of course expected to facilitate pacemaking. L- and T-type $Ca^{2+}$ channels are known to play an important role in pacemaking: they are thought to contribute to the diastolic depolarisation underlying pacemaking and be responsible for the action potential upstroke in the centre of the sinus node[52]. The auxiliary $Ca^{2+}$ channel subunits, $Ca_v\alpha_2\delta_1$/ Cacna2d1 and $Ca_v\alpha_2\delta_2$/Cacna2d2, were more abundant in the sinus node. There is strong evidence that the surface membrane expression of $Ca_v1$ channels is increased by $\alpha_2\delta$ subunits[53]. $Ca_v\alpha_2\delta_2$/Cacna2d2 increases T-type $Ca^{2+}$ current amplitude[54], and has previously been reported to be confined to the nodal regions in the mouse[49]. The T-type $Ca^{2+}$ channel, $Ca_v3.2$/Cacna1h, was more abundant in the sinus node as expected[55], but surprisingly $Ca_v3.1$/Cacna1g was not detected in our proteomics data albeit we measured transcripts from it in our scRNAseq data. In summary, the data are consistent with more abundant expression of $Ca^{2+}$ channels in the sinus node, which is expected to facilitate pacemaking. The differential expression of plasma membrane ion channels stands in contrast to the uniform expression of $Ca^{2+}$ clock proteins. Together with our modelling results, we conclude that the observed differences in expression of ion channels between the sinus node and atrial muscle can explain the pacemaker activity of the sinus node.

Our analysis showed that the metabolic profile of the sinus node is associated with synthesis, accumulation and storage of lipids. We find this is in part explained by nodal cells containing lipid droplets. Lipid droplets can contribute significantly to mitochondrial fatty acid supply and take part in a wide range of cellular processes, including modulation of protein availability, fatty acid trafficking and lipid signalling[56], as well as protection against lipid toxicity[57]. Yet the over-accumulation of lipid droplets is associated with disease, including heart failure and dilated cardiomyopathy[58]. In this respect, the sinus node may be particularly vulnerable to lipotoxicity in a disease setting. Several of the lipogenic proteins we found in our proteomics analysis are predominantly or exclusively associated with adipocytes. pointing to the sinus node being surrounded by cardiac adipose tissue, which is consistent with previous studies[59,60]. This hypothesis is confirmed by the observation in our snRNA-seq data that the majority of transcripts related to synthesis, accumulation and storage of lipids are predominantly expressed in the adipocyte cell clusters. The finding of overexpression of UCP1 further supports this contention, as cardiac adipose tissue is known to express thermogenic genes uniquely associated with brown adipose tissue[61,62]. Energy storage is particularly important, as lipids are the primary fuel source for myocytes; therefore, the close proximity of epicardial adipose tissue allows for rapid lipid mobilisation, which supports the high energy demands of myocytes.

The sinus node is characterised by a high content of extracellular matrix (ECM)[29]. It is likely that the ECM provides mechanical support for the sinus node, and in addition, the ECM is proposed to isolate pacemaker myocytes from the hyperpolarizing influence of the surrounding atrial myocytes[24,63,64]. A distinctive feature of the proteomic datasets was the presence of high levels of ECM glycoproteins and proteoglycans. Most significantly, sinus node ECM was enriched for components of elastic microfibrils, a finding supported by subsequent immunohistochemical analysis. The presence of high levels of elastin and elastic microfibrils suggests that these components endow the tissue with above average recoil. It is conceivable that a higher level of elastic ECM components in the sinus node contributes to pacemaker function given the role of integrin–ECM ligand

interactions in mechanosensing[65–67], presenting potential synergy between adhesion and stress-activated channels. Sinus node ECM also possesses a characteristic composition of collagens. Of the 28 different collagen types, our analyses indicate that only types VI, XII, XV and XVIII are enriched in sinus node. Mutations in collagen type VI and XII cause Bethlem myopathy in man[68–71], characterised by contractures of multiple joints and a generalised skeletal muscular wasting and weakness, while defects in types XII, XV and XVIII (the latter pair are homologues of each other) have been associated with other forms of muscle degeneration and cardiomyopathy in mice[72–74]. Taken together, these studies point to a key role for certain collagens in maintaining the tissue structure required for efficient muscle function, including in the heart, and suggest that alterations in their abundance or distribution may contribute to heart conditions. The role of aberrant extracellular matrix in the sinus node is worthy of further study, because fibrosis is thought to be a cause of sinus node dysfunction by interrupting connections between sinus node cells. See Supplementary Discussion for further elaboration on the membrane and $Ca^{2+}$ clocks, ion channels, kinases, contractile proteins, natriuretic peptide system, transcription factors (Supplementary Fig. 21), ECM and lipid metabolism.

In summary, the data-driven proteomics approach elucidates both the long-standing debate on sinus node pacemaking, as well as previously unsuspected characteristics of the sinus node tissue. The data reveal specialisations of the sinus node, including metabolism, ion cycling and extracellular matrix; aberrations of which are all strongly implicated in cardiac disease. Changes in sinus node function as well as sinus node dysfunction is implicated in a wide range of circumstances ranging from pregnancy and ageing to disease states such as heart failure and myocardial infarction[1,75]. The prospective ability to study remodelling of the sinus node proteome, as well its cellular composition, in disease states has the potential to transform our molecular understanding of diseases linked to sinus node dysfunction and to open the way to the development of new therapeutic strategies.

## Methods

**Tissue collection**. Thirty male C57BL/6J mice, 12–14-weeks-old and 25–35 g body weight were consecutively euthanized by cervical dislocation in accordance with the United Kingdom Animals (Scientific Procedures) Act, 1986 and complied with Directive 2010/63/EU of the European Parliament. The care and usage of animals was according to standards and practices approved by the University of Manchester Animal Welfare and Ethical Review body and in compliance with the Animals (Scientific procedures) Act, 1986. The hearts were quickly removed and the right atrium with the whole of the sinus node (Fig. 1b) was dissected out in cold phosphate-buffered saline (Sigma-Aldrich, UK). For the sinus node, the translucent intercaval region between the crista terminalis and the atrial septum (~1 by 1 mm) was dissected. In separate cohort of hearts, tissue sections through the translucent intercaval region were immunolabelled for hyperpolarization-activated cyclic nucleotide-gated channel (HCN4) and connexin 43 (Cx43) proteins using protocols reported previously[76] and described below in the immunohistochemistry section. The translucent intercaval region labelled positive for HCN4 and negative for Cx43, typical of the sinus node (Fig. 1b). For right atria, part of the right atrial appendage (~1 by 1 mm) was dissected. Tissue samples were placed in cryotubes and snap-frozen in liquid nitrogen. Samples from ten animals were pooled per tube resulting in three tubes each for sinus node and atrial muscle. Pooling of samples was necessitated by the small size of the mouse sinus node and the need to collect sufficient protein for detection by mass spectrometry. Tissue samples were shipped on dry ice and stored at −80 °C until use. The subsequent experiments and data analysis were performed blinded.

**Tissue homogenisation**. Frozen tissue biopsies were homogenised on a Precellys24 homogeniser (Bertin Technologies, France) in a Tris-based triton x-100 buffer (50 mM Tris-HCl pH 8.5, 5 mM EDTA, 150 mM NaCl, 10 mM KCl, 1% Triton X-100, 5 mM NaF, 5 mM beta-glycerophosphate, 1 mM Na-orthovanadate, containing 1x Roche complete protease inhibitor) with ceramic beads (2.8 and 1.4mm zirconium oxide beads, Precellys). Homogenates were incubated for 2 h at 4 °C (20 rpm) and subsequently transferred to chilled 1.5 mL tubes. Tissue lysates were sonicated for five cycles (30 s on/30s off at low amplitude) in a Bioruptor

sonicator (Diagenode, USA), cleared by centrifugation (15,000 × g, 20 min, 4 °C), and soluble fractions collected. To remove surfactants prior to LC-MS/MS measurement, protein was precipitated by addition of trichloroacetic acid (TCA) to a final concentration of 10%. Samples were incubated for 30 min on ice and centrifuged gently (200 × g, 20 s) to harvest protein. Supernatants were discarded and protein resuspended in Guanidine-HCl buffer (Gnd-HCl; 6MGnd-HCl, 50 mM Tris-HCl pH 8.5, 5 mM NaF, 5 mM beta-glycerophosphate, 1 mM Na-orthovanadate, containing 1x Roche complete protease inhibitor). Disulfide bridges were reduced and cysteine moieties alkylated by addition of 5 mM Tris(2-carboxyethyl) phosphine (TCEP) and 10 mM chloroacetamide (CAA) and incubation in the dark at room temperature for 15 min.

**Peptide preparation**. From each sample, up to 1 mg protein was digested in-solution by addition of endoproteinase Lys-C (Trichem ApS, Denmark; 1:100 enzyme:protein ratio) for 1 h at 30 °C, 750 rpm in the dark, followed by dilution (1:12 with 50 mM Tris-HCl pH 8) and digestion with trypsin overnight (16 h) at 37 °C, 750 rpm (Life technologies, USA, 1:100 enzyme:protein ratio). The reactions were quenched by addition of trifluoroacetic acid (TFA, 1% final conc.) and samples were centrifuged (14,000 × g, 10 min) to sediment debris. Soluble fractions were desalted and concentrated on C18 SepPak columns (Waters, USA). Peptides were eluted with 40% acetonitrile (ACN) followed by 60% ACN, and organic solvents subsequently evaporated by vacuum centrifugation.

**Offline high pH fractionation of peptide samples**. Of each sample, 100 μg peptide (in 10 μL injection volume) was fractionated by micro-flow reverse-phase ultra-high pressure liquid chromatography (UPLC) on an Dionex UltiMate 3000 UPLC system (Thermo Scientific, USA) equipped with an ACQUITY UPLC CSH C18 Column (130Å, 1.7 μm, 1 mm x 150 mm) at 30 μL/min flow rate[6,7]. The following 85 min gradient elution programme was used employing a binary pump connected to Solvent A (5 mM ammonium bicarbonate (ABC), pH 8) and B (100% ACN): 0–50 min: 4.5–22.5% B, 50–55 min: 22.5–63% B, 55–60 min: 63% B isocratic, 60–62 min: 63–81% B, 62–70 min: 81% B isocratic, 70–75 min: 81–4.5% B, followed by column re-equilibration at 4.5% B for 10 min. Outflow from 4–60 min was collected in 1-min intervals into 12 concatenated fractions in the autosampler. Fractions were acidified by addition of 5 μL 5% formic acid (FA) to avoid precipitation of ABC, and fraction volume was reduced by vacuum centrifugation. If not specified otherwise, chemicals and reagents were acquired from Sigma-Aldrich, USA. Chromatography solvents were acquired from VWR, USA.

**LC-MS/MS measurements**. Fractionated peptide samples were analysed by online reversed-phase liquid chromatography coupled to a Q-Exactive HF quadrupole Orbitrap tandem mass spectrometer (Thermo Electron, Bremen, Germnay). Peptide samples were brought to concentration of 0.2 μg/μL (diluted in 5% ACN, 0.1% TFA) in 96-well microtiter plates and autosampled (5 μL injection volume) into a nanoflow Easy-nLC system (Proxeon Biosystems, Odense, Denmark). Peptide samples were separated on 15 cm fused-silica emitter columns (75 μm inner diameter) pulled and packed in-house with reversed-phase ReproSil-Pur C18-AQ 1.9 μm resin (Dr. Maisch GmbH, Ammerbuch-Entringen, Germany) in a 1 h multi-step linear gradient (0.1% formic acid constant; 2–25%ACN in 45 min, 25–45% ACN in 8 min, 45–80%ACN in 3 min) followed by short column re-equilibration (80–5%ACN in 5 min, 5%ACN for 2 min). Column effluent was directly ionised in a nano-electrospray ionisation source operated in positive ionisation mode and electrosprayed into the mass spectrometer. Full-MS spectra (375–1500 m/z) were acquired after accumulation of 3,000,000 ions in the Orbitrap (maximum fill time 25 ms) at 120,000 resolution. A data-dependent Top12 method enabling deep proteome measurements[8] was used, sequentially isolating the most intense precursor ions (up to 12 per full scan) for higher-energy collisional dissociation (HCD) in an octopole collision cell. MS/MS spectra of fragment ions were recorded at resolution of 30,000 after accumulation of 100,000 ions in the Orbitrap (maximum fill time 45 ms).

**Raw data processing**. Raw MS data was processed using the MaxQuant software v1.5.3.30[77] and proteins identified with the built-in Andromeda search engine by searching MS/MS spectra against an in silico tryptic digest of a database containing all reviewed SwissProt protein entries (downloaded on 6.8.2016). The MS/MS spectra were searched with Carbamidomethly Cysteine as fixed modification, as well as oxidation (M), acetylation of protein N-termini and Gln->pyro-Glu as variable modifications. A maximum of two missed cleavages and six variable modifications was allowed. The minimum peptide length was set to seven amino acids (default) and minimum Andromeda score required for modified peptides was 25, with minimum delta score of 6 (default). First search tolerance was 20 ppm (default) and main search tolerance was 4.5 ppm (default), requiring strict specificity of tryptic peptides. Owing to the similarity of the samples, the match-between-runs option was enabled with default parameters. False-discovery rate cutoffs were set to 1% on peptide, protein and site decoy level (default), only allowing high-quality identifications to pass. All peptides were used for protein quantification, and label-free quantification (LFQ) was additionally performed in MaxQuant with fast LFQ option enabled. Protein identification results were further processed using the Perseus software suite[78] version 1.5.5.3 and the R

Programming Environment (R v3.3.3 in RStudio v1.0.136). Quantile normalisation of raw intensities was performed based on the Bioconductor R package LIMMA[79]. For quantitative analysis, protein groups representing isoforms of the same canonical protein were removed from the data, such that only the most abundant protein isoform was retained.

**Quantitative analysis**. Global analysis of significant differences between sinus node (SN) and right atrial (RA) samples was performed in Perseus on LFQ values of technical replicate A due to better overall data quality of this replicate. Reverse and contaminant identifications were removed from the data and isoform information was collapsed into a canonical protein group (i.e., if more than one isoform was identified only the isoform with highest intensity was considered). Principal component analysis (PCA) was performed on 100% valid values. For all other quantitative analyses, data was filtered so that only proteins with at least three valid values in at least one of the cardiac regions (SN or RA) were retained.

**Volcano plot analysis**. For global analysis of quantitative differences (volcano plot), missing values were imputed separately for each column from the lower end of the original. Specifically, imputation of missing values was performed in the Perseus software with standard settings: values were drawn from a distribution with width 0.3 and downshift 1.8 compared to the original sample distribution. Imputation was performed separately for each sample. The 5523 proteins remaining after filtering resulted in 33,138 theoretical values ($6 \times 5523$) of which 32,433 were valid (measured) and the remaining 819 were imputed (~2.5% of total). The maximum number of imputed values for one sample was 194 values (SN3) (SN1), followed by SN3 (190), SN2 (178), RA1 (94), RA2 (82) and RA3 (81 values). Only proteins identified with at least two peptides were included in the subsequent analysis. Perseus volcano plot analysis was performed applying two-sided Student's $t$-test to test for significant difference of protein abundance between SN and RA. A permutation-based false-discovery rate (FDR) cutoff was determined with 750 randomisations and $S_0 = 0.1$ (default). Out of 5523 proteins remaining after filtering, 587 were deemed significant at 5% FDR. Among the proteins with imputed values, we subsequently manually checked significant proteins to ensure that significance was never based on imputation due to poor quantification. To this end, we examined raw intensities of all significant proteins that only had 2, 1 or zero LFQ values in either SN or RA. If we had recorded raw intensities, but they had not been quantified in LFQ, we double-checked whether significance was only due to loss of data points during quantification (and subsequent imputation), i.e., if the ratio of raw intensities of LFQ intensities between SN and RA strongly deviated from the ratio observed after imputation. On this basis we decided to exclude a list of 12 proteins from further analyses, considered very likely to be false positives due to strongly overestimated fold-change difference (SN/RA) when compared to raw intensities: Cyb5d2, Fxyd1, Gfra2, Ca4, Ntm, Mup3, Sema7a, Ddx18, Fgd4, Gas2l1, Ptpn18 and Cend1.

**Network analysis**. In order to investigate associations between significant proteins, we extracted protein associations from the STRING database[80] for all significant proteins from Volcano Plot analysis. STRING identified 566/575 significant proteins and returned 2176 interactions at confidence cutoff 0.4 (medium confidence level in STRING). No interactions were returned for 91 of the proteins, whereas 463 clustered into a main network and the remaining ten into smaller detached networks of 2–3 proteins. We performed MCL clustering on the network in Cytoscape[81] using Clustermaker2[82]. The clustering was performed relative to score at inflation factor 3.0, so that higher scoring interactions were given more weight during clustering. Out of 109 clusters returned, 24 clusters contained five or more members (encompassing 228 proteins in total). Ontology enrichment analysis was performed for clusters with five or more members using the STRING App (v1.1.0) in Cytoscape[80,81]. From the enrichment analysis, overall terms descriptive of major differences between sinus node and right atrial clusters were deduced. For analysis of smaller groups of functionally related proteins, we extracted protein associations for human proteins from STRING and overlaid mouse protein expression data. Confidence level was set to medium confidence (0.4) in all networks except the action potential network; here the score cutoff was lowered to 0.3 in order to include th HCN4-HCN1 association.

**Immunohistochemistry**. Right atrial tissue preparations with intact sinus node ($n = 2$; typical preparation shown in Fig. 1b) were embedded in OCT compound (VWR International, UK) and frozen by pouring 2-methylbutane (Sigma-Aldrich, UK) cooled in liquid $N_2$. Frozen serial sections (20 μm thickness) were cut perpendicular to the crista terminalis from the proximal to the distal end of the preparations in a Leica CM3050 S cryostat (Leica Microsystems, Germany) at −18 °C. The cut sections were placed onto SuperFrost Plus slides (VWR International, UK) and stored at −80 °C until further use. Frozen serial sections were fixed in 10% formalin made up in phosphate buffered saline (PBS; Sigma-Aldrich, UK) for 30 min and later washed with PBS (Sigma-Aldrich, UK) three times (10 min/wash). Tissues were permeabilised with 0.1% Triton X-100 in PBS (Sigma-Aldrich, UK) for 30 min, washed three times in PBS (10 min/wash) and blocked with 1% bovine serum albumin (BSA; Sigma-Aldrich, UK) made in PBS for ~30 min. The BSA was removed and tissue sections incubated at 4 °C overnight with appropriate primary

antibodies diluted in 1% BSA in PBS. All antibodies used in this study were commercially available, validated by suppliers for species reactivity and for immunofluorescence applications and are listed in Supplementary Table 1. Supplier recommended dilutions were used. Following overnight incubation with primary antibodies, tissue section were washed three times in PBS (10 min/wash) and incubated for 2 h at room temperature with appropriate secondary antibodies (see Table with all antibody information in Supplementary Material) diluted in 1% BSA in PBS. Post-incubation with the secondary antibodies, sections were washed three times with PBS (10 min/wash). A drop of Vectashield antifade mounting medium (H-1000, vector laboratories, USA) was placed on tissue sections and cover slips were attached to slides with nail varnish. Immunofluorescence labelling of proteins was detected by laser scanning confocal microscopy (LSM5, Carl Zeiss, Germany) using LSM5 Zeiss Pascal software (Carl Zeiss, Germany) for image acquisition. Laser light wavelengths and filters appropriate for the secondary antibody fluorophores were used. Three tissue sections from each right atrial preparation were used for each antibody and four images were obtained from each tissue section. Representative images of antibody labelling are shown in Figs. 1b, 4 and Supplementary Fig. 17.

**Intracellular action potential recording**. Sinus node and right atrial action potentials were recorded using sharp microelectrodes in spontaneously beating tissue preparations (Fig. 1b). Tyrode solution (contents in mM: NaCl 120.3, KCl 4.0, $CaCl_2$ 1.2, $MgSO_4$ 1.3, $NaH_2PO_4$ 1.2, $NaHCO_3$ 25.2 and glucose 11) bubbled with 95%$O_2$/5%$CO_2$ mixture to maintain pH 7.4 was circulated at 20 mL/min and tissues were maintained at 37 °C. The centre of the sinus node was mapped with a set of two bipolar extracellular electrodes. Using 3 M KCl filled sharp microelectrodes of 20–40 MΩ tip resistance multiple records of intracellular action potentials were obtained at the sinus node centre and pectinate muscle (for right atrial tissue). These recording sites were consistent with the tissue biopsy sites for MS. The voltage difference across the plasma membrane was acquired at 20 kHz, passed through a 10 kHz low-pass Bessel filter and amplified 10x in Axon Instruments GeneClamp 500 amplifier (Molecular Devices, USA). Data was digitised in Axon Instruments Digidata 1440A (Molecular Devices, USA) and stored onto a computer for generating action potential plots (Fig. 3 insets).

**Conversion of an atrial to SN-like action potential model**. Can pacemaking in the sinus node be explained by the differences in ion channel expression in the sinus node? To answer this question, a biophysically detailed model of the mouse atrial action potential[14] was modified based on the mass spectrometry data. First, the cell capacitance was reduced from 50 to 25 pF (typical for atrial and sinus node cells, respectively). Label-free quantified intensity data were used to scale ionic current conductances for $I_{Na}$, $I_{Ca,L}$ and $I_{K,r}$ (Supplementary Table 2). A different approach had to be used for $I_f$ and $I_{Ca,T}$, because they are not present in the mouse atrial action potential model. Equations for $I_f$ and $I_{Ca,T}$ were taken from an independent model of the mouse sinus node action potential[15] and introduced into the mouse atrial action potential model; for a mouse atrial action potential the ionic conductances for the two currents were reduced in line with the mass spectrometry data; when simulating the sinus node action potential, the ionic conductances for the two currents were not reduced. In addition, the half-activation voltage of $I_{Ca,T}$ was shifted by −7 mV as compared to the formation for $I_{Ca,T}$ from the mouse sinus node action potential model from which the formation was taken[15]. $I_{K,s}$ is not in the mouse atrial action potential model and, therefore, it was not considered. The ion channels ($K_v1.5$ and $K_v4.x$) responsible for $I_{K,ur}$ and $I_{to}$ were not detected. Nevertheless, the ionic conductances for $I_{K,ur}$ and $I_{to}$ had to be reduced to 0% and 10%, respectively, because the presence of these currents in the sinus node resulted in a triangular shaped action potential; this must be considered a limitation of the modelling, because it is a change in the absence of justification from mass spectrometry. The ion channel normally considered responsible for $I_{K,1}$ (Kir2.x) was not detected by mass spectrometry. It is possible that Kir3.x or TASK1 channels are responsible for it instead and both ion channels were less abundant in the sinus node (Supplementary Table 2). The majority of simulations were carried out assuming that $I_{K,1}$ was unchanged in the sinus node. However, in some simulations the ionic conductance for $I_{K,1}$ was reduced in line with the observed changes in Kir3.1 and TASK1 channels; this increased the pacemaking rate but otherwise had little effect (data not shown). Sarcoplasmic reticulum $Ca^{2+}$ uptake and release were scaled in line with the mass spectrometry data (Supplementary Table 2). In simulations, the atrial action potential model was paced at a cycle length of 200 ms. All cell models were run for 20 s to reach a steady state and the last two action potentials were recorded for analysis. Characteristics of the simulated action potentials are shown in Supplementary Table 3.

We also did the reverse. We used the model of Kharche et al.[15] of the mouse sinus node cell action potential as a starting point. The model was modified to include discrete ionic currents carried by the different HCN channels. The model was then modified in the same way as above, but in reverse. To successfully convert from the sinus node action potential to an atrial-like action potential (non-spontaneous, triangular action potential shape), we had in addition to make a change in the background inward rectifier $K^+$ current ($I_{K,1}$). As the mass spectrometry provided no data on Kir2 channels (ion channels thought to be responsible for $I_{K,1}$), $I_{K,1}$ from the model of the mouse atrial cell action potential was incorporated. This resulted in the successful conversion of the sinus node

action potential to an atrial-like action potential. Given the lack of proteomic evidence for Kir2, one could speculate whether it is possible that not Kir2 channels are responsible for $I_{K,1}$, but instead TASK1 and/or Kir3 channels may be responsible, which do change in the expected manner if they are responsible for $I_{K,1}$.

**Calculation of ion channel numbers in the sinus node**. The mass spectrometry data were used to estimate the relative number of different ion channels in a sinus node myocyte. To convert the relative numbers of ion channels to an absolute number of ion channels in a sinus node myocyte, an independent estimate of the absolute number of an ion channel in a sinus node myocyte is required. A Markov chain model can replicate the behaviour of a single ion channel. By summing the ionic current for multiple ion channels (all of the same type), the number of ion channels required to replicate whole cell current can be estimated. Markov chain models for five single ion channels (Hcn1, Hcn4, $Ca_v1.2$, $Ca_v3.2$ and ERG) were developed based on the Wang et al.[83] model of a mouse sinus node action potential. These Markov chain models considered an open and closed state for Hcn1 and Hcn4 and two additional inactive states for $Ca_v1.2$, $Ca_v3.2$ and ERG (Supplementary Fig. 9). The stochastic behaviour of the ion channels was simulated by using a Monte Carlo method mimicking the transition between states. The unitary channel conductances used are listed in Supplementary Table 4. In order to verify stochastic opening and closing of the ion channels, we simulated consecutive sweeps of the unitary ion channel models during a predefined voltage clamp protocol (Fig. 5 and Supplementary Figs. 8–11). Finally, the number of each ion channel required to account for the whole cell current was calculated. The calculated number of HCN4 ion channels per sinus node myocyte was used to convert the relative numbers of ion channels in a sinus node myocyte (based on mass spectrometry) to an absolute number of ion channels in a sinus node myocyte. Models of action potentials were numerically solved by a differential equation solver CVODE[84]. In simulations, a time step of 0.1 ms was used. The stochastic behaviour of single channel models was mimicked by the Monte Carlo method with a smaller time step of 0.02 ms. We calculated the number of ion channels in SN myocytes based on their intensity-based absolute quantification (iBAQ)[12]. For each channel protein we searched the literature to find the number of subunits in a functional channel (e.g., four for HCN4 and one for $Ca_v1.2$) and divided the iBAQ intensity with the number of subunits constituting a channel (Supplementary Table 3). We then used our estimate of 6255 HCN4 channels per SN myocyte derived from Markov chain modelling to convert iBAQ intensities to copy numbers per cell. From these data we calculated a mean and a standard error of the mean, which are plotted as bar graphs (Fig. 5 and Supplementary Fig. 7).

**Electron microscopy**. Sinus node and right atrial tissue samples ($\sim 1 \times 1$ mm) were collected as described above for mass spectrometry. The protocol from Starborg et al.[85] was used to prepare and image tissue samples for electron microscopy. Specifically, freshly collected samples were fixed in 2.5% glutaraldehyde in 100 mM cacodylate buffer (pH 7.2) for at least 1 h and stored at 4 °C until further use. Tissues were washed three times in distilled water (5 min/wash) and placed in a freshly made mixture of 1% osmium tetroxide and 1.5% potassium ferrocyanide in 0.1 M cacodylate buffer for 1–2 h. Subsequently, tissues were washed three times in distilled water (5 min/wash) and transferred into freshly made 1% tannic acid in 0.1 M cacodylate buffer for 1–2 h followed by five distilled water washes (5 min/wash). It is vital that tannic acid is thoroughly washed off. Samples were incubated in 1% osmium tetroxide in double distilled water for 40 min at room temperature and further washed in distilled water (three washes, 5 min/wash) before incubation in aqueous 1% uranyl acetate for at least 1 h and stored overnight at 4 °C. On the following day, samples were washed three times in distilled water (5 min/wash) and dehydrated by subsequent exchanges of the following ethanol dilutions in distilled water: 30, 50, 70, 90 and two times 100% for 15 min at each step at room temperature. This was followed by two changes of acetone, 30 min each. Samples were infiltrated in graded series of TAAB 812 Hard (TAAB Laboratories Equipment Ltd, UK) resin in acetone at room temperature: first overnight in 25% TAAB 812, then 8 h in 50% TAAB 812, and then overnight in 75% TAAB 812 and finally 6 h in 100% TAAB. Samples were transferred to fresh 100% TAAB 812 Hard in labelled moulds and allowed to cure at 60 °C for 24–48 h. Small pieces of resin embedded tissues were mounted on aluminium specimen pins (Gatan, USA) using cyanoacrylate glue. The blocks were faced and precision trimmed with a diamond knife to a square ($\sim 0.5 \times 0.5$ mm) so that tissue was exposed on all four sides and prepared for examination with a Gatan 3view microtome (Gatan, USA) within an FEI Quanta 250 FEG (Thermo Fisher Scientific, UK) serial block-face scanning electron. Serial images of the block-face were captured at an accelerating voltage of 3.8 kV, a spot size of 2.8 and pressure of 1.6 Torr. The dwell time for each micrograph was 4 µs. Pixel size was 24.4 × 24.4 nm (XxY) and Z-slice thickness was 75 nm. The horizontal and vertical field width were ~100 and ~200 µm, respectively. For transmission electron microscopy (TEM) sections were cut with Reichert Ultracut ultramicrotome and observed with FEI Tecnai 12 BioTWIN microscope (FEI Company, USA) at 100 kV accelerating voltage. Images were taken with Gatan Orius SC1000 CCD camera and presented in Supplementary Figs. 14–16.

**Proteome measurements of fibroblasts**. Hearts were explanted from three male mice, washed in ice-cold phosphate buffered saline (PBS) and ventricles dissected on ice. The ventricles were finely cut with a scalpel and digested using 2.5 mL digestion cocktail (Miltenyi Biotech) followed by trituration. Isolation DMEM [high glucose, glutamax, no pyruvate, 100 µm ascorbic acid, 20% fetal bovine serum (FBS, Gibco) 0–9% Penicillin/Streptomycin (P/S, Gibco)] were added to each sample followed by filtration through a 70 µm cell strainer. Filtrates were centrifuged at $500 \times g$ for 5 min, the cell pellet was resuspended in 1 mL PEB-buffer (phosphate buffered saline (PBS), pH 7.2, 0.5% bovine serum albumin (BSA), 2 mM EDTA). 10 mL 1x red blood cell lysis solution (Miltenyi Biotech) was added and the samples centrifuged at $600 \times g$ for 5 min. The cell pellets were each resuspended in PBS containing 15 µL enzyme A (Miltenyi Biotech) and centrifuged at $600 \times g$ for 5 min. The cell pellets were resuspended in [high glucose, glutamax, no pyruvate, 100 µm ascorbic acid, 9% FBS, 0–9% P/S] and seeded on poly-D-lysine coated dishes. After 2 h at 37 °C in a humidified incubator with 5% CO2, the cells were washed twice, detached and counted using the viability and cell count assay on a Nucleocounter 3000 (Chemometec). Cells were spun down at $10,000 \times g$ for 7 min and after removal of the supernatant frozen at −80 °C. For peptide preparation, the three fibroblast samples were processed separately. First, the cells were lysed by adding 20 µL of a 1% TritonX-100 solution (1x proteinase inhibitor complete (Roche), 50 mM β-glycerophophosphate, 10 mM sodium orthovanadate, 5 mM magnesium chloride in PBS). One microgram of DNase was added and the samples were incubated on ice for 1 h while shaking. 70 µg of beads (1:1 stock of Sera-Mag Carboxylate-Modified Magnetic Particles (hydrophylic, 24152105050250) & Sera-Mag Carboxylate-Modified Magnetic Particles (hydrophobic 44152105050250 by GE Healthcare) were added per sample and ethanol added to a final of 70%. After 10 min at room temperature (RT), samples were placed on a magnet for 1 min, beads were washed twice with 70% ethanol and then resuspended in 20 µL 50 mM HEPES pH 8.5. TCEP was added to a final concentration of 5 mM and CAA to a final concentration of 5.5 mM. After 30 min at RT, 0.25 µg lys-C (Trichem ApS, Denmark) were added and samples incubated at 37 °C for 1 h. Furthermore, 0.5 µg trypsin (Life technologies, USA) were added and samples digested overnight at 37 °C. The peptides were separated from the beads by applying a magnetic field, transferred to a new tube and acidified to 1% final Trifluoroacetic acid (TFA). Peptides were cleaned using $C_{18}$ SepPak cartridges (Waters). Peptides were eluted using 40% acetonitrile, containing 0.05% TFA. Before MS measurement, the desalted peptide mixture was separated into 12 concatenated fractions on a Dionex UltiMate 3000 UPLC system (Thermo Scientific, USA) as described above[6,7]. Fractionated peptide samples were injected on a 15 cm column (75 µM inner diameter) packed with 1.9 µM C18 beads (Dr. Maisch GmbH) using an Easy-LC 1200 (Thermo Fisher Scientific) and separated on a 1 h linear gradient with increasing buffer B (80% acetonitrile, 0.1% formic acid) at a flow rate of 250 nL/min. All samples were analysed on a Q-Exactive HF-X (Thermo Fisher Scientific) mass spectrometer operated in positive, data-dependent mode using a Top12 method to enable deep proteome measurements (MS1 resolution = 60.000, MS2 resolution = 15.000). Raw MS data was processed using the MaxQuant software v1.6.1.0[77] using default settings and a database containing all reviewed SwissProt protein entries (downloaded on 26 July 2018).

**Single-nucleus RNA sequencing**. Sinus nodes were isolated from 16 mice as described above. Sinus nodes from four animals were pooled, snap-frozen in liquid nitrogen and stored at −80 °C until processing. For nuclei isolation, samples were kept on dry ice prior to addition of 1 mL Nuclei EZ Lysis Buffer (Sigma-Aldrich). The tissue was triturated by pipetting 10x with a 1000 µL tip, followed by 10x with a 200 µL tip. The samples were incubated on ice for 5 min, with tube inversion every minute. After another trituration round (10x 1000 µL tip), a centrifugation at $800 \times g$ for 7 min was carried out. The supernatant was carefully removed and the nuclei and remaining tissue pieces were resuspended in 500 µL resuspension buffer (0.1% BSA containing 0.2 U/µL Protector RNase inhibitor). The samples were filtered through a 70 µm nylon mesh that had been soaked in RNAse away overnight and washed in MilliQ. Afterwards the filter was washed with 500 µL resuspension buffer and the samples were centrifuged at $800 \times g$ for 7 min. Nine-hundred ninety microlitres of the supernatant was removed and 10 µL resuspension buffer added. The isolated nuclei were visually inspected and loaded into two channels on the Chromium system (10x genomics) using manufactures protocol (Chromium Single-Cell 3' Reagent Kits (v3 Chemistry)) to obtain Single-nucleus libraries that were sequenced on a NovaSeq™ 6000 sequncing system (Illumina) using a S1 Reagent Kit (100 cycles). For single-nucleus RNA sequencing read alignment, the Cellranger 3.0 pipeline (10x Genomics, USA) was used to align reads, quantify unique molecular identifiers (UMI) and generate feature-barcode expression matrices. The reads were mapped to an adapted reference from 10x mm10-premrna to include introns according to the procedure set out on the 10x Genomics website [https://support.10xgenomics.com/single-cell-gene-expression/software/pipelines/latest/advanced/references]. The raw expression matrices produced by the Cellranger count tool were subsequently analysed using the Seurat v.3.0.0.9000 R package[86]. In line with 10x Genomics indications, for each sample, based on an estimated 9000 cells loaded, the 5157 barcodes with the highest number of counts were initially selected for downstream analysis [https://kb.10xgenomics.com/hc/en-us/articles/360001378811-What-is-the-maximum-number-of-cells-that-can-be-profiled-]. In a further filtering step, based on visual

inspection of QC plots, barcodes with at least 3000 UMI and proportions of mitochondrial and ribosomal RNA below 0.08 and 0.1, respectively, were retained. The DoubletFinder R package[87] was used to remove suspected doublets. Of the original 5127 barcodes selected from each channel, 4426 nuclei from the first channel and 931 from the second channel survived the QC, with the mitochrondrial and ribosomal expression ceilings accounting for the majority of the omitted barcodes. The data was lognormalized with the LogNormalize() command, the FindVariableFeatures() command was used to find the top 2000 variable genes and the ScaleData() command was used to scale the expression data to Z-scores and regress out confounding variation associated with the number of UMI or with ribosomal or mitochondrial gene content. The RunPCA() command was used to compute 75 principal components (PCs). The FindNeighbors() and FindClusters() commands were used with default parameters to identify cell clusters, and the FindMarkers() command was used with the MAST method from the MAST R package[88] (v1.8.2, https://github.com/RGLab/MAST/) to identify differentially expressed genes. These genes were used to manually add cell type annotations to the clusters. The RunTSNE() command was used to a perplexity of 30 to produce a 2-D T-distributed stochastic neighbour embedding (t-SNE). Clusters and gene expression panels were visualised using the ggplot2 R package[89].

**Reporting summary**. Further information on research design is available in the Nature Research Reporting Summary linked to this article.

## Data availability
All MS raw data and search results from this study were uploaded to the ProteomeXchange Consortium via the PRIDE repository[90] with the identifier PXD008736. The single-nucleus RNA transcriptomic sequencing data discussed in this publication have been deposited in NCBI's Gene Expression Omnibus[91] and are accessible through GEO Series accession number GSE130710. The data underlying Figs. 1, 2, 4, 5, 6, 7, 9 and Supplementary Figs. 1, 4, 7, 13, 17, 18, 20, 21 are provided in Supplementary Data 1–7. Source Data for Figs. 5a, 6b and Supplementary Figs. 4c, 13c, 18b, 21 are provided as Source Data file. A reporting summary for this Article is available as a Supplementary Information file. All other data supporting the findings of this study are available from the corresponding authors on reasonable request.

## Code availability
The custom code to reproduce single-nucleus RNA transcriptomic sequencing data analysis has been deposited at https://github.com/perslab/linscheid-2019.

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

## Acknowledgements

Supported by grants from the Carlsberg Foundation (CF17-0209) and The Danish Council for independent Research DFF-4092-00045 and DFF-6110-00166 to A.L., the Lundbeck Foundation to T.H.P. (R19020143904) and the British Heart foundation (RG/18/2/33392) to M.B. The Wellcome Trust Centre for Cell-Matrix Research, University of Manchester, is supported by core funding from the Wellcome Trust (203128/Z/16/Z). The Novo Nordisk Foundation Centre for Basic Metabolic Research is partially funded by an unconditional donation from the Novo Nordisk Foundation. The Novo Nordisk Foundation Centre for Protein Research is funded in part by a generous donation from the Novo Nordisk Foundation (NNF14CC0001). We would like to thank the staff of the Wellcome Centre for Cell Matrix Research Electron Microscope facility in the Faculty of Biology Medicine and Health, University of Manchester for their support. In particular, we acknowledge the support of staff members David J. Smith, Aleksandr Mironov and Tobias Starborg, as well as Andrew J. Atkinson, Abimbola Akerele and Tharsika Sri-tharan. We would furthermore like to thank the PRO-MS Danish National Mass Spectrometry Platform for Functional Proteomics and the CPR Mass Spectrometry Platform for instrument support and assistance. We would further like to thank Alex-ander Hogrebe for input on figure graphics. Some elements in Figs. 1a and 2b were downloaded and reused from Servier Medical Art [https://smart.servier.com/] by Servier licensed under a Creative Commons Attribution 3.0 Unported License [https://creativecommons.org/licences/by/3.0/].

## Author contributions

N.L.: sample preparation, mass spectrometry measurements, all bioinformatic data analyses of proteomics data, writing and approval of the manuscript. S.J.L.: collection of tissue biopsies, all electrophysiology, immunohistochemistry and electron microscopy, writing and approval of the manuscript. P.C.P.: sample preparation, mass spectrometry measurements, approval of the manuscript. M.S.: fibroblast and nuclei sample preparations. S.Z.: calculation of ion channel numbers based on Markov chain modelling, computer modelling of the sinus node action potential, writing and approval of the manuscript. G.G.: interpretation of metabolism data, writing and approval of the manuscript. M.J.H.: interpretation of extracellular matrix data, writing and approval of the manuscript. K.L.E.: single-cell RNA sequencing measurements, approval of manuscript. J.J.T.: analysis of single-cell RNA sequencing data, approval of manuscript. T.P.: supervision of single-cell RNA sequencing experiments, approval of manuscript. A.K.: contribution to electron microscopy, approval of the manuscript. H.Z.: calculation of ion channel numbers based on Markov chain modelling, computer modelling of the sinus node action potential, writing and approval of the manuscript. J.V.O.: contribution to mass spectrometry methods and data evaluation, approval of the manuscript. M.B.: conception of the project, acquisition of funding, interpretation of data, writing and approval of the manuscript. A.L.: conception of the project, acquisition of funding, supervision of all aspects of mass spectrometry data acquisition and data analysis, interpretation of data, writing and approval of the manuscript.

## Additional information

**Competing interests:** The authors declare no competing interests.

