## [Peer Review File · Nature Communications]

Reviewers' comments:

Reviewer #1 (Remarks to the Author):

This is a very well thought through and carried out scientific work. This work is of great quality and was aimed at performing a quantitative proteomic analysis of the sinus node to decipher molecular mechanisms associated with cardiac pacemaking. The work is the first comprehensive proteome investigation to date of the sinus node and it is very well written. Authors did not just simply limit themselves to just performing proteomic analysis without a thorough interpretation of findings, they backed their results with a thorough pathway analysis actually revealing pathways of relevance. Several additional scientific experiments were conducted and provided ample evidence to support the claims made by the authors. The supplementary files provide a detailed information about the work carried and also an additional discussion. Data uploaded on PRIDE were checked and the MS raw files were indeed uploaded as mentioned in the manuscript.

Only minor comments I would like to point out:

- What would be the clinical benefit/impact of the study? This can be shortly mentioned in the conclusion.
- How can these findings in mice be translated in humans especially knowing the genetic differences between humans and mice? This is further highlighted by the lack of suitable mouse models totally reflecting human heart diseases.

Reviewer #2 (Remarks to the Author):

The paper by Linscheid et al is a very strong and interesting manuscript reporting novel findings in the mouse heart. It takes a current proteomic approach to investigate unique regional proteomic differences in the mouse heart, with careful detail to the analysis. The sample size (pooled fractions of dissected regions in 3 separate experiments) appears to be appropriate with 144 separate MS runs. Pooling all of their mass spec data, they report the identification of over 7000 potential proteins in the study.

Experimentally, the authors focus on membrane proteins that they identified to have differential expression between the atria and the 'sinus node'. They have biochemical and confocal assays

confirming that established pacemaker ion channel family members (Hyperpolarization-activated Cyclic Nucleotide-gated channels- HCN 1,4) are enriched in their appropriate fraction.

Experimental electrophysiology (fig 3) shows isolated single Sinus nodal cells vs atrial cell action potentials. Similarly, EM experiments (fig 8) highlight the general architecture of this known domain.

In this manuscript, the authors attempt to quantify the 'amount' or copy number of the ion channels in the membranes. I found this approach to provide significant novelty to the field and will be of sufficient general interest and is a unique aspect of this paper. Modeling experiments attempt then to correlate their copy numbers to modelled action potentials, which are consistent with experimentally known data.

The network analysis employs the usual informatic approaches; string network analysis is provided. This is sufficient, although not very informative. The data highlights that additional potentially differential protein groups are likely associated with the different regions in the heart.

Concerns

1. While not a necessity, how does this study compare with the recent papers, such as the by the Doll et al., NComm paper (Mann group) who performed a similar comprehensive analysis of HUMAN heart. Although human nodal cells were not enriched in the Mann study, they have reported a similarly comprehensive atrial proteome, along with a model of electrical disease (atrial fibrillation). A common set of proteins or datasets would be valuable.
2. Dissection of the regions in the heart is likely not the best approach to isolate such a unique population of nodal cells. There are reporter strains, allowing for flow sorting of acutely dissociated nodal cells. Such an approach would allow for a very precise analysis of this population through directed enrichment, or at the last a depletion of other cell populations. Similarly, although not perfect, nodal cell cultures have been reported in the literature, and at least would provide a very unique population to confirm this proteome. One manuscript revision may be to similarly assess the proteome of mouse or human nodal derived populations; and compare to the existing in vivo 'nodal' data. This would be valuable to stem cell biologists trying to engineer nodal cells, and /or highlight the differences in an in vivo region.
3. Why is so much of the paper restricted to the membrane protein analysis. Other cellular characteristics are likely involved in differential function between the nodal vs atrial myocyte, including metabolism, structural, or similar. This analysis was presented in a restricted manner with discussion in the Supplement, but it could be presented in a clearer format (other than just the String pathway figure) within the main text.

4. I do not believe that the channel number calculations took into account the heterogeneity of the cell population from a biological perspective or take that aspect into account in their specific calculations. So, maybe calculations can be made on the proportion of cell types represented in each sample, or change/modify the claims that these are myocyte numbers. Maybe the differences between the SN and atrial tissue are due to differing proportions of the cell types in each sample. At the least, it needs to be discussed/rationalized.

Additional issues:

1. I was not aware of the ongoing 'long lasting debate' of the pacemaker. This was clearly settled with the identification and study of the pacemaker currents as the primary effector of the pacing of the nodal cell.

2. How exactly was the nodal region dissected? This is a VERY small region and is represented by probably less than 1% of the cells in the heart. Importantly, this region is not clearly delineated by gross morphology assessments. As the authors do show in their histology, it is also comprised of many different cell types, including myocytes, fibroblasts, endothelial cells and perhaps some inflammatory cells. Many cellular approaches will use some sort of reporter mouse, either b-gal positive, etc. to allow for this region to be reproducibly identified and the margins defined. Given the tiny size of the area, this is perhaps the most critical aspect of this study to be clear on. A 'perfect' experiment would be to flow sort out cell populations and do the proteomics on acutely dissociated cells.

3. In the supplemental methods the authors mention that some data imputation was performed, but there is no mention on how many values needed to be imputed. Similarly, whether it was done across technical replicates? This can be clarified.

4. Lastly the authors do not provide the quantitative values for the atrial tissue in their supplementary table of all proteins, just the 'SN' number. This may be an issue of just not labeled properly since all intensity columns are labeled as "SN".

5. Spelling of 'atrial' (ATRIL) in supplementary dataset - all proteins. Supp Table 1

Reviewer #3 (Remarks to the Author):

Nora Linscheid and colleagues present a quantitative proteomic analysis of the mouse sinoatrial node region and the right atrial tissue. Over 7000 proteins were quantified in both tissues, and 575 proteins were found to be differently expressed. They then focus on a few specific aspects of the profile, the implications for pacemaking mechanism by membrane and/or calcium clock, ion channel/ion handling protein expression and abundance in general, and extracellular matrix.

The proteomic map is interesting. However, the study is descriptive, and the novelty beyond the proteomic map itself is limited.

Comments

1) The source tissues were taken by biopsies. The tissue composition of the biopsies is a concern. Sinus node and atrium are composed of different ratios of distinct cell types (cardiomyocytes, different types of fibroblasts and endothelial (derived) cells, epicardial cells, endocardial cells, smooth muscle cells, nerve fibers, blood cells). This will affect the quantification. Moreover, the source cell type of each protein cannot be defined. Therefore, particular processes (e.g. metabolic activity) cannot be attributed to pacemaker cells. A preselection or pre-purification of particular cell types (e.g. cardiac muscle cells, if that is considered an interesting cell type for the electrophysiological properties of the sinus node) would circumvent these issues.

2) While the atrial tissue is unlikely to be contaminated by sinus node tissue, the sinus node tissue, small and irregular, is likely to be contaminated by atrial tissue when taking biopsies. Consistently, the differences in level of known atrial-enriched proteins between sinus node tissue and atrial tissue is very small. Recent RNAseq gene expression profiling (Vedantham 2015) of purified mouse sinus node regions and atrial tissue revealed much greater differences in expression of Cx43, Cx40, Nppa, Nppb, Kcnj3, etc than reported here. Interestingly, sinus node enrichment of Hcn4 is similar between the published RNA profiles and proteomic analysis presented here (>100 fold) suggesting the discrepancies do not result from mRNA vs protein quantification. This needs to be addressed as the current quantifications may be biased to sinus nodal tissue-derived protein enrichment.

3) The quantification should be compared to the various RNA-seq/expression profiling data sets of sinus nodal and atrial tissues. Many of the proteins or level differences reported here as novel can be found in these data sets. Moreover, mRNA and protein level comparisons may provide insight into discrepancies, which then can be tested.

4) While the computer modeling is not within my field of expertise, I have the impression that it does not provide novel insights. The key currents of the sinus node and their differences with those of the atrial working myocytes are well established, and the model recapitulates that. Interventions to address the role of the different components and mechanisms would be required to gain novel

insight. Several of these interventions (e.g. Hcn4 knock down) have already been published. Moreover, extrapolating channel activity from protein abundance ignores important post-translational modifications, localization, etc. involved in channel activity and currents.

Reviewer #4 (Remarks to the Author):

I only have minor comments related to the modeling. The choice of model (Shen et al) should be justified. There are well-established alternatives, why was this model used?

The gain of automaticity after changing the current density to sinus node expression levels is certainly an argument for the membrane-clock. It would strengthen the argument if one could also show the opposite effect, i.e. modifying a model of the sinus node to atrial myocyte expression levels would induce a loss of automaticity.

Response to Reviewers on “Quantitative proteomic map of the sinus node elucidates the foundation of cardiac pacemaking”

Reviewer #1 (Remarks to the Author):

This is a very well thought through and carried out scientific work. This work is of great quality and was aimed at performing a quantitative proteomic analysis of the sinus node to decipher molecular mechanisms associated with cardiac pacemaking. The work is the first comprehensive proteome investigation to date of the sinus node and it is very well written. Authors did not just simply limit themselves to just performing proteomic analysis without a thorough interpretation of findings, they backed their results with a thorough pathway analysis actually revealing pathways of relevance. Several additional scientific experiments were conducted and provided ample evidence to support the claims made by the authors. The supplementary files provide a detailed information about the work carried and also an additional discussion. Data uploaded on PRIDE were checked and the MS raw files were indeed uploaded as mentioned in the manuscript.

We would like to thank the reviewer for thoroughly evaluating our work, and not least we would like to express our gratitude for the explicit appraisal of our work. We find this to be rare, and it is greatly appreciated.

Only minor comments I would like to point out:

- What would be the clinical benefit/impact of the study? This can be shortly mentioned in the conclusion.

We appreciate the suggestion from the reviewer and have added a section on the potential clinical impact of the study.

The following is added to the Discussion section on page 10 (blue type):

“Changes in sinus node function as well as sinus node dysfunction is implicated in a wide range of circumstances ranging from pregnancy and ageing to disease states such as heart failure and myocardial infarction^{1, 2}. The prospective ability to study remodeling of the sinus node proteome, as well its cellular composition, in disease states has the potential to transform our molecular understanding of diseases linked to sinus node dysfunction and to open the way to the development of new therapeutic strategies”

- How can these findings in mice be translated in humans especially knowing the genetic differences between humans and mice? This is further highlighted by the lack of suitable mouse models totally reflecting human heart diseases.

The reviewer is correct that translating findings from animal models to humans is a general challenge. We have expanded on the translational perspective of our findings by adding a section discussing the limitations of our study.

On page 13-14 of the Supplementary Material we have added the following paragraph:

“The study we present here was conducted on mice. Available evidence suggests that study of the sinus node of laboratory animals (e.g. mouse, rat, rabbit) helps in understanding the human sinus node. Pacemaker ionic currents and the pacemaker transcriptome of the mouse and human sinus nodes are similar and animal models of many different conditions (pregnancy, postnatal development, ageing, from day to night, athletic training, heart failure, myocardial infarction, metabolic syndrome, diabetes) replicate the same sinus node phenotype as observed in humans^{1, 3-5}.”

Reviewer #2 (Remarks to the Author):

The paper by Linscheid et al is a very strong and interesting manuscript reporting novel findings in the mouse heart. It takes a current proteomic approach to investigate unique regional proteomic differences in the mouse heart, with careful detail to the analysis. The sample size (pooled fractions of dissected regions in 3 separate experiments) appears to be appropriate with 144 separate MS runs. Pooling all of their mass spec data, they report the identification of over 7000 potential proteins in the study.

Experimentally, the authors focus on membrane proteins that they identified to have differential expression between the atria and the ‘sinus node’. They have biochemical and confocal assays confirming that established pacemaker ion channel family members (Hyperpolarization-activated Cyclic Nucleotide-gated channels- HCN 1,4) are enriched in their appropriate fraction.

Experimental electrophysiology (fig 3) shows isolated single Sinus nodal cells vs atrial cell action potentials. Similarly, EM experiments (fig 8) highlight the general architecture of this known domain.

In this manuscript, the authors attempt to quantify the ‘amount’ or copy number of the ion channels in the membranes. I found this approach to provide significant novelty to the field and will be of sufficient general interest and is a unique aspect of this paper. Modeling experiments attempt then to correlate their copy numbers to modelled action potentials, which are consistent with experimentally known data.

The network analysis employs the usual informatic approaches; string network analysis is provided. This is sufficient, although not very informative. The data highlights that additional potentially differential protein group are likely associated with the different regions in the heart.

We would like to thank the reviewer for his/her expert insights in the evaluation of our study, for his/her general positive evaluation of our work, and for appreciating the significant novelty in our analyses on ion channel copy numbers.

Concerns

1. While not a necessity, how does this study compare with the recent papers, such as the by the Doll et al., NComm paper (Mann group) who performed a similar comprehensive analysis of HUMAN heart. Although human nodal cells were not enriched in the Mann study, they have reported a similarly comprehensive atrial proteome, along with a model of electrical disease (atrial fibrillation). A common set of proteins or datasets would be valuable.

We agree with the reviewer that it is important to evaluate datasets in light of previously published datasets. However, the focus of our study is on the sinus node, which is a unique region of the heart. As the reviewer also writes, the sinus node is not covered in the dataset by Doll and colleagues. That is, the part of our dataset that makes our dataset unique can not be compared to previous publications. What we can do, is to compare the atrial proteome we measured for mice with the proteome Doll *et al.* measured for human heart. This analysis is primarily a comparison between what is conserved between humans and mice.

To compare the atrial proteome published by Doll *et al.*⁶ with our mouse proteome, we first mapped human Uniprot identifiers to mouse Uniprot identifiers through BioMart. We then queried the human cardiac right atrial proteome data for homologs to each of the proteins we quantified in mouse in our study. For more than 80% of the mouse right atrial proteins we have quantified, a human ortholog was mapped in the human right atrial proteome. That is, even with the challenges presented by mapping between species, we find that more than 80% of the mouse right atrial proteome is also expressed in human right atria. The next question is how well the protein abundances between mouse right atria and human right atria correlate. As evidenced in the figure below, correlating protein abundance in the right atria of mouse and human revealed a correlation of 0.73 between the datasets. This result is remarkable considering the differences in sample handling and MS measurements between our mouse-based study and the human study by Doll and colleagues. The correlation between datasets of 0.73 indicates that the right atrial proteomes of the two species resemble each other. This is an important observation from a translational point of view. Yet, care must be taken to assess the suitability of mouse models for each given research question individually.

We have added the following text on page 14 in the Supplementary Material:

“For the atrial proteomes we have the possibility to compare the similarities between the murine right atrial proteome we measured here with human right atrial proteomes that were recently published⁶. Mapping human Uniprot identifiers to mouse Uniprot identifiers through BioMart enabled us to find a human ortholog in the human right atrial proteome for more than 80% of the mouse right atrial proteins we quantified in this study. Evaluating the protein abundances in murine right atria to protein abundances in human right atria present a correlation of 0.73. This indicates that the right atrial proteomes of the two species resemble each other. This is an important observation from a translational point of view. Yet, it is important to stress that the data presented in this study are based on mice and care must be taken to assess the suitability of mouse models for each given research question individually.”

And we have added to page 8 of the manuscript:

“It is the only proteome of the sinus node, which makes it an important addition to other current cardiac protein atlases of the heart⁶.”

2. Dissection of the regions in the heart is likely not the best approach to isolate such a unique population of nodal cells. There are reporter strains, allowing for flow sorting of acutely dissociated nodal cells. Such an approach would allow for a very precise analysis of this population through directed enrichment, or at the last a depletion of other cell populations. Similarly, although not perfect, nodal cell cultures have been reported in the literature, and at least would provide a very unique population to confirm this proteome. One manuscript revision may be to similarly assess the proteome of mouse or human nodal derived populations; and compare to the existing in vivo ‘nodal’ data. This would be valuable to stem cell biologists trying to engineer nodal cells, and /or highlight the differences in an in vivo region.

We would like to thank the reviewer for his/her constructive suggestions.

We also agree with the reviewer that it would indeed be interesting to isolate primary nodal cells and investigate proteomes of isolated cell populations. Unfortunately, this is not currently possible. Whereas millions of ventricular or atrial myocytes can be isolated from a heart, from the mouse heart we can isolate ~300 sinus node cells. With current state-of-the-art technology, it is not feasible to measure proteomes of a deep and quantitative quality based on so few cells.

The reviewer suggests that we could consider to compare our results to results that we could obtain from a cell line. Baruscotti and DiFrancesco have developed and published on a stem cell derived sinus node cell line, which we are collaborating with them on. Another example is the commercially available 'Shox2' stem cell derived sinus node cell line. It is not known how accurate a sinus node cell line these are and a study of the proteome of a cell line is in our opinion premature. The sinus node-like cell lines are by no means perfect, and as their similarities and differences to endogenous sinus node cells are not known in detail, we would not know exactly what we are comparing our data to if we were to perform the suggested experiment. In our opinion, the cell lines are less relevant to study than the endogenous tissue. It is possible to reverse the argument presented: our study of the proteome of the natural sinus node will be a reference point for the development of a stem cell derived sinus node cell line in the future.

We do agree with the reviewer that these are important points to discuss in our paper. For that reason we have added the following section to the 'Limitations of the study' paragraph on page 14 of the Supplementary Material:

"Furthermore, our work is based on analysis of biopsy samples. Although this is the standard in fields measuring cardiac proteomes or transcriptomes, it does represent a challenge. Work on biopsies suffer the general limitation that tissue is composed of different cell types, as we also explicitly show in our study. Cell sorting of acutely dissociated cells offers a potential solution to this problem. However, whereas millions of ventricular or atrial myocytes can be isolated from a heart, from the mouse heart we can isolate ~300 sinus node cells. With current state-of-the-art technology it is not feasible to measure proteomes of a deep and quantitative quality based on so few cells. Another possibility would be to use a sinus node-like cell line, but it is not known how faithful such cell lines are and a study of the proteome of a cell line would be premature. Conversely, our study of the proteome of the natural sinus node will be a reference point for any sinus node cell line.

In this study, we did not use a molecular reporter to identify the sinus node and take a biopsy, contrary to what others have done⁷. Instead the site of biopsy was guided by our previous studies of the anatomy of the sinus node using histology or micro-CT, immunolabeling of marker proteins such as HCN4 to confirm the nature of the tissue, mapping the electrical activity of multicellular sinus node preparations (for example using extracellular potential recording) to identify the leading pacemaker site, and recording of intracellular action potentials from in and around the leading pacemaker site with sharp microelectrodes⁸. Based on our years of experience from these studies, we know where the leading pacemaker site is located in the mouse heart and we took a biopsy at this location: at the bifurcation of the sinus node artery in the intercaval region towards the superior vena cava."

3. Why is so much of the paper restricted to the membrane protein analysis. Other cellular characteristics are likely involved in differential function between the nodal vs atrial myocyte, including metabolism, structural, or similar. This analysis was presented in a restricted manner with discussion in the Supplement, but it could be presented in a clearer format (other than just the String pathway figure) within the main text.

We agree with the reviewer and we are pleased that the reviewer recognises that there are many interesting findings in our dataset besides the ones related to membrane proteins. The reason why we placed a substantial amount of discussion of our findings in the Supplementary Material is the space limitation set by the journal. However, we agree with the reviewer that there are several findings that deserve further mention. For the same reason, we have performed additional electron microscopy experiments, which allow us to better discuss the findings.

We have added Supplementary Figs. S14-S16, supplementary methods on p. 7 of the Supplementary Information, and Supplementary Videos 1 and 2. Furthermore, we have added new text on page 6-7 of the paper:

“Consistent with this, electron micrographs showed perinuclear lipid vesicles in sinus node myocytes (and atrial myocyte; Supplementary Figs. S14c, S15g) as well as adipocytes in the sinus node. The sinus node is characterised by a haphazard array of small (5-10 μm) spindle-shaped sinus node myocytes, but other cell types are also present (Fig. 8a) and they leave their signature in the proteomics data. Neurofilament proteins (forming intermediate filaments in autonomic axons⁹) were more highly expressed in the sinus node (Fig. 7). Consistent with this, electron micrographs revealed a high abundance of autonomic nerve fibers in the sinus node and few in the atrial muscle (Fig. 8a-b; Supplementary Fig. S14a,e and S15c). Electron micrographs showed that in the sinus node (and atrial muscle) the epicardium is lined with numerous finger-like cilia (Fig. 8a-b; Supplementary Fig. S15a-b). Kinesin II (Kif3)¹⁰ and tubulin (Tuba)¹¹ are essential constituents of cilia, and the presence of cilia is reflected in measurements of Kif3a, Kif3b, Kif3c, Tuba1a, Tuba1b, Tuba1c, Tuba3a, Tuba4a and Tuba8 in the sinus node proteome (Supplementary Table S1). Macrophages have been reported in the atrioventricular node and shown to be functionally important for conduction¹² and macrophages are evident in the electron micrographs of the sinus node (Fig. 8a; Supplementary Fig. 14g). Consistent with this, CD163, a marker of M2 macrophages^{12, 13} was present in the sinus node (Supplementary Table S1). In addition, the sinus node artery (with associated cell types), telocytes (novel type of interstitial cell reported in the human sinus node¹⁴), monocytes and fibroblasts were observed in the electron micrographs of the sinus node (Supplementary Fig. S14g). The fibroblasts help in the formation of the extracellular matrix and the electron micrographs showed that the sinus node cells (unlike atrial myocytes) are loosely packed within copious extracellular matrix including basement membrane, elastin and wavy bundles of collagen fibers (Fig. 8a; Supplementary Fig. S16).”

4. I do not believe that the channel number calculations took into account the heterogeneity of the cell population from a biological perspective or take that aspect into account in their specific calculations. So, maybe calculations can be made on the proportion of cell types represented in each sample, or change/modify the claims that these are myocyte numbers. Maybe the differences between the SN and atrial tissue are due to differing proportions of the cell types in each sample. At the least, it needs to be discussed/rationalized.

We were pleased to read that the reviewer found our approach to estimate ion channel copy numbers a significant novelty and a unique aspect of our paper. To acknowledge this point mentioned by the reviewer in the general statement on our study, we have adjusted our abstract such that this finding is also mentioned in the abstract.

Our evaluations of ion channel copy numbers per myocyte are exclusively made for sinus node tissue, and hence removes the concern of different cellular compositions of atria and sinus node tissues.

We do specifically state that the calculation is based on the assumption that the contribution to measured channel abundance only comes from myocytes and not from other cell types. We wrote “Based on our measurements, the estimated channel copy numbers will rely on the assumption that the ion channel measurements originate from myocytes and the majority of the subunits reside in the plasma membrane” (p. 5).

We do agree with the reviewer that our argument would be stronger if this statement is backed by evidence. Therefore, we decided to evaluate experimentally if substantial contribution to ion channel abundance is caused by other cell types. In our paper we have shown evidence from electron micrographs that the sinus node contains many different cell types. However, there is no doubt that the most prevalent cell type of the sinus node is myocytes. Myocytes by far make up the majority of the volume of the sinus node. The second most abundant cell type in the sinus node is fibroblasts. We hence reasoned that to evaluate if another cell type could significantly contribute to ion channel abundance, the contribution should come from fibroblasts.

As is the case for sinus node myocytes, it is also not possible to study fibroblasts isolated from the sinus node itself. Instead, we acutely isolated fibroblasts from mouse ventricles and treated these isolated cells essentially the same way as we treated the biopsy samples to perform proteome measurements of the cardiac fibroblasts. It is important to note that we did not grow these cells in culture. We made this choice because fibroblasts are known to change their molecular phenotype during culturing. From this primary cell isolation we were able to measure a proteome from the fibroblasts at a depth of 5,518 proteins, which is a great depth for differentiated cells. We identified 20 different ion channel proteins in the fibroblasts. We correlated the measured protein intensities in the fibroblasts with those measured in the sinus node proteome (see panel a, figure below). Overall, there protein intensities measured in the sinus node proteomes and those measured in the fibroblast proteome are well correlated with a correlation coefficient $R=0.63$. In the correlation plot we have highlighted the abundances of the ion channels found in both datasets in orange, and the figure shows that their abundances in

the two datasets are similar. The ion channel abundances are further highlighted in the rank sum plot below (panel b+c). Again this figure highlights the similarities between the abundances of ion channels measured in the sinus node proteome with the abundances measured in the fibroblast proteomes, for those channels that we identified in the fibroblasts.

For this matter, the most important observation is that none of the plasmalemmal channels of focus in Figure 2 was identified in the fibroblasts (panel b+c). That is, the voltage gated ion channels involved in the membrane clock are not measured in the deep fibroblast proteome. With the limitation in mind that these fibroblasts are isolated from heart ventricle we conclude that the potential contribution to ion channel abundance from fibroblasts is minimal.

Comparison of SN tissue proteome with proteome of isolated cardiac fibroblasts. a. Correlation between protein intensities in both datasets was $r=0.63$. All ion channels identified in both datasets are highlighted. **b.+c.** Rank plots of protein identifications for both datasets. Ion channels are highlighted as identified in each of the datasets, showing that most membrane ion channels (yellow) were not identified in fibroblasts.

Based on our findings from proteome measurements of acutely isolated fibroblasts we evaluate that the likely contamination of ion channel abundance from other cell types is a minor risk factor. To further support this argument, we also performed single cell RNA sequencing experiments on sinus node biopsies. The snRNA-seq data also confirmed that transcripts from the ion channels in question were exclusively expressed in the sinus node myocytes. Details are provided in the response to reviewer #3. Yet, we would like to stress that the results we present on ion channel copy numbers per cell represent a first attempt to estimate these.

To expand on the concern raised by the reviewer, we have added the following to our manuscript:

On page 5 of the manuscript:

“Is it reasonable to assume that ion channel abundances originate from myocytes? Currently, we cannot measure deep proteomes of isolated sinus node myocytes to evaluate this hypothesis due to the limited number of cells. We reasoned that the only other cardiac cell type

that could substantially contribute to channel abundance would be fibroblasts, due to their high abundance in the heart. Thus to query for ion channel protein abundance in cardiac fibroblasts we isolated primary cardiac fibroblasts and measured a proteome at a depth of >5,500 proteins (details provided in the Supplementary Material). Only 20 channels were measured in fibroblasts (Supplementary Fig. S5). Importantly, none of the channels of focus here were among these 20 channels. Hence, we consider the assumption that ion channel abundances originate from myocytes to be reasonable.

On page 8 of the manuscript:

“For instance, our ion channel copy numbers estimate relied on the assumption that the channels investigated are exclusively expressed in the sinus node myocytes. From the snRNA-seq data we extracted information on which sinus node cell types express the ion channel mRNA and found that the membrane clock proteins were indeed predominantly expressed in the sinus node myocytes (Supplementary Fig. S20).”

We have added the following sentence to our abstract to highlight this finding of the paper:

“Combining our quantitative proteomics data with computational modeling, we estimate ion channel copy numbers for sinus node myocytes.”

We have added a detailed corresponding methods section to the Supplementary Material p.8-9.

Additional issues:

1. I was not aware of the ongoing ‘long lasting debate’ of the pacemaker. This was clearly settled with the identification and study of the pacemaker currents as the primary effector of the pacing of the nodal cell.

There has been a debate ever since the first recordings were made of ionic currents in the sinus node and the debate continues to this day. First, there was the debate whether the pacemaker potential is generated by the deactivation of outward current or activation of inward current, then there was the debate whether it is generated by the funny current or background inward current, and currently there is the ongoing debate whether it is generated by the membrane clock or the calcium clock¹⁵. There is also a variety of other ion channels (e.g. Cav3 channels) said to be involved in pacemaking. Our study shows that HCN4 (the mainstay of the membrane clock) is the most differentially expressed protein between the sinus node and atrial muscle, whereas calcium clock proteins are not differentially expressed and this is new and important evidence in favour of the membrane clock as we state in the paper.

2. How exactly was the nodal region dissected? This is a VERY small region and is represented by probably less than 1% of the cells in the heart. Importantly, this region is not clearly delineated by gross morphology assessments. As the authors do show in their histology, it is also comprised of many different cell types, including myocytes, fibroblasts, endothelial cells and perhaps some inflammatory cells. Many cellular approaches will use some sort of reporter mouse, either b-gal positive, etc. to allow for this region to be reproducibly identified and the

margins defined. Given the tiny size of the area, this is perhaps the most critical aspect of this study to be clear on. A 'perfect' experiment would be to flow sort out cell populations and do the proteomics on acutely dissociated cells.

Please see our response to your 'Concern 2' above where we address these important issues.

Additionally, we compared mRNA sequencing data from samples dissected by an identical approach as we used in this study (unpublished data) to mRNA data from a study employing a reporter mouse strain (Vedantham *et al.*⁷). As shown in the figure below mRNA profiles between the two different approaches are reasonably similar, considering that different mouse strains and sample preparation workflows were employed (correlation coefficient $R=0.62$ between datasets).

Comparison of mRNA sequencing results in sinus node from laser microdissection versus tissue biopsy protocol. Two-dimensional density distribution of mRNA sequencing data from Vedantham *et al.* (2015) obtained from laser microdissection of a reporter mouse strain (x axis) versus mRNA sequencing data of our tissue isolation protocol (y axis) shows correlation of 0.62 and centering around identity line (light-blue line; light blue part of distributions contains highest point density). Analysis based on 11022 matching data points between datasets. CPMR: counts per million reads.

This figure has been added as new Supplementary Figure S2 to the Supplementary Information. Additionally, we have added to following sentence to our manuscript on page 3:

"In a previous study a different strategy was applied to isolate the sinus node, which was subsequently subjected to transcriptomics analysis. Both isolation strategies achieve similar

results (Supplementary Fig. S2). We find the transcriptomics and proteomics data to be medium correlated (Supplementary Fig. S3), which is expected, and supports the notion of complementarity of the two sets of information^{16, 17}

3. In the supplemental methods the authors mention that some data imputation was performed, but there is no mention on how many values needed to be imputed. Similarly, whether it was done across technical replicates? This can be clarified.

We agree with the reviewer that it is important to be specific on how imputation has been performed and not least to what extend data has been imputed. We have certainly made sure to provide this information in the manuscript, but it may not have been sufficiently detailed.

Imputation of missing values was performed in the Perseus software with standard settings: Values were drawn from a distribution with width 0.3 and downshift 1.8 compared to the original sample distribution. Imputation was performed separately for each sample. The 5523 proteins remaining after filtering resulted in 33138 theoretical values (6×5523) of which 32433 were valid (measured) and the remaining 819 were imputed (~2.5% of total). The maximum number of imputed values for one sample was 194 values (SN1), followed by SN3 (190), SN2 (178), RA1 (94), RA2 (82) and RA3 (81 values). The portion of imputed values is further illustrated in the figure below in red.

We have added the following section to page 4-5 of the Supplementary Material:

“Specifically, imputation of missing values was performed in the Perseus software with standard settings: Values were drawn from a distribution with width 0.3 and downshift 1.8 compared to the original sample distribution. Imputation was performed separately for each sample. The 5523 proteins remaining after filtering resulted in 33138 theoretical values (6*5523) of which 32433 were valid (measured) and the remaining 819 were imputed (~2.5% of total). The maximum number of imputed values for one sample was 194 values (SN1), followed by SN3 (190), SN2 (178), RA1 (94), RA2 (82) and RA3 (81 values).”

4. Lastly the authors do not provide the quantitative values for the atrial tissue in their supplementary table of all proteins, just the ‘SN’ number. This may be an issue of just not labeled properly since all intensity columns are labeled as "SN".

We would like to thank the reviewer for noticing this mistake, which is now corrected.

5. Spelling of ‘atrial’ (ATRIL) in supplementary dataset - all proteins. Supp Table 1

We would like to thank the reviewer for noticing this mistake, which is now corrected.

Reviewer #3 (Remarks to the Author):

Nora Linscheid and colleagues present a quantitative proteomic analysis of the mouse sinoatrial node region and the right atrial tissue. Over 7000 proteins were quantified in both tissues, and 575 proteins were found to be differently expressed. They then focus on a few specific aspects of the profile, the implications for pacemaking mechanism by membrane and/or calcium clock, ion channel /ion handling protein expression and abundance in general, and extracellular matrix.

The proteomic map is interesting. However, the study is descriptive, and the novelty beyond the proteomic map itself is limited.

We would like to thank the reviewer for his/her efforts in making an in-depth evaluation of our manuscript. We disagree with the reviewer that our study merely is descriptive. Our manuscript provides a large resource of information on the sinus node protein composition, as well as on its cellular composition with this revised version. The amount of information contained in the dataset as whole surpasses what can be evaluated in detail in a single paper. For detailed analysis, we have focused on one aspect of the dataset, namely the proteins argued to be important for pacemaking. Our proteomics data suggest that the membrane clock proteins are essential for pacemaking function. And we confirm our finding by an orthogonal approach based on computational modeling. Furthermore, based on our data we make a first attempt to actually quantify ion channel copy numbers per myocyte. Fortunately, other reviewers appreciated the significance and novelty of these parts of our work.

Comments

1) The source tissues were taken by biopsies. The tissue composition of the biopsies is a concern. Sinus node and atrium are composed of different ratios of distinct cell types (cardiomyocytes, different types of fibroblasts and endothelial (derived) cells, epicardial cells, endocardial cells, smooth muscle cells, nerve fibers, blood cells). This will affect the quantification. Moreover, the source cell type of each protein cannot be defined. Therefore, particular processes (e.g. metabolic activity) cannot be attributed to pacemaker cells. A preselection or pre-purification of particular cell types (e.g. cardiac muscle cells, if that is considered an interesting cell type for the electrophysiological properties of the sinus node) would circumvent these issues.

We agree with the reviewer that it would be ideal to study individual cell populations isolated from the tissue regions of interest, if this could be achieved in a manner where the state of the isolated cells were not affected and where sufficient amounts of cells could be isolated. However, for mouse sinus node this is not possible with current state-of-the-art techniques. The majority of studies on cardiac transcriptomes and proteomes have been carried out using cardiac biopsies as in this study. Work on biopsies suffers the important limitation that tissue is composed of different cell types. Flow-sorting of acutely dissociated cells offers a potential solution to this problem. If the ventricular or atrial myocardium was being studied, it would be possible to flow-sort acutely dissociated working myocytes. Many millions of myocytes can be isolated from the ventricles. In contrast, from the mouse heart we can typically isolate ~300 sinus node cells. It is not feasible to measure proteomes of a deep and quantitative quality based on so few cells, even if samples were pooled.

To tackle the challenge of assigning protein findings to particular cell types, we decided to pursue single cell RNA sequencing experiments. Combining our protein based findings with information on the cell types actually expressing the protein under study would allow us to interpret the protein findings in the correct cellular context.

We write in the manuscript on pages 7-8:

“Single-nucleus RNA sequencing of sinus node tissue elucidates the cellular diversity and pinpoints protein differences to specific cell types”

From our quantitative proteomics analysis of the sinus node and atrial tissues we have identified specific protein differences between these tissues. As evidenced by the electron microscopy data presented above, it is challenging to attribute the protein differences to a particular cell type due to the cellular complexity of the tissue. To overcome this challenge, we performed snRNA-seq of mouse sinus node biopsies. Specifically, sinus node tissue was isolated in a similar manner as described for the proteomics experiments, and for each replicate, sinus nodes from four mice were pooled. We extracted nuclei from the biopsies and prepared single-nuclei suspensions for snRNA-seq analysis. We chose this preparation protocol to minimize transcriptional profile alterations during sample handling. We obtained single-nucleus libraries using the Chromium system (10X Genomics) and sequenced them on a Illumina NovaSeq 6000. After quality control, the dataset comprised 5357 nuclei transcriptomes and covered a total of 27998 genes, with a per cell average of 2715 expressed genes. The dataset of 27998 transcripts are provided in Supplementary Table S6. This data enabled us to undertake a

comprehensive transcriptomics characterization of the mouse sinus node. When projecting the nuclei to a two-dimensional t-distributed stochastic neighbor embedding (tSNE) plot, we observed a clear distinction between cell types of the sinus node identifying 12 distinct cell clusters (Fig. 9a). As expected, the clusters comprised sinus node myocytes and fibroblasts. We also identified macrophages, endothelial and endocardial cells and epithelial and epicardial cells. Interestingly, we also found 3 cell clusters with gene profiles suggesting subpopulations of adipocytes expressing marker genes indicating differing levels of thermogenic activity (Fig. 9b). Our snRNA-seq data illustrates the cellular complexity of the sinus node tissue beyond what could be inferred from the electron micrographs. Importantly, we can utilize the snRNA-seq data in the interpretation of our protein based findings. For instance, our ion channel copy numbers estimate relied on the assumption that the channels investigated are exclusively expressed in the sinus node myocytes. From the snRNA-seq data we extracted information on which sinus node cell types express the ion channel mRNA and found that the membrane clock proteins were indeed predominantly expressed in the sinus node myocytes (Supplementary Fig. S20). For any protein discussed in this paper, the snRNA-seq dataset is provided as a reference point to identify the cell population expressing the protein (Supplementary Tables S6 and S7).”

Figure 9: Single-nucleus RNA sequencing analysis of sinus node tissue reveals the major cell types and their transcript profiles. a. Two-dimensional t-SNE plot outlines the major cell populations of the sinus node. Each point represents a single nucleus. Cell populations are colored according to cluster designation. b. Centered and scaled expression levels of cell cluster markers for sinus node myocytes, fibroblasts, macrophages, vascular endothelial cells, endocardial cells, epicardial cells, adipocytes, and neurons.

The experimental procedure of the work is added to the Supplementary Material on p.9.

The limitation of working with biopsy material is now covered in a section on the limitations of our study, Supplementary Material on page 14:

“Furthermore, our work is based on analysis of biopsy samples. Although this is the standard in fields measuring cardiac proteomes or transcriptomes, it does represent a challenge. Work on biopsies suffer the general limitation that tissue is composed of different cell types, as we also explicitly show in our study. Cell sorting of acutely dissociated cells offers a potential solution to this problem. However, whereas millions of ventricular or atrial myocytes can be isolated from a heart, from the mouse heart we can isolate ~300 sinus node cells. With current state-of-the-art technology it is not feasible to measure proteomes of a deep and quantitative quality based on so few cells. Another possibility would be to use a sinus node-like cell line, but it is not known how faithful such cell lines are and a study of the proteome of a cell line would be premature. Conversely, our study of the proteome of the natural sinus node will be a reference point for any sinus node cell line.”

2) While the atrial tissue is unlikely to be contaminated by sinus node tissue, the sinus node tissue, small and irregular, is likely to be contaminated by atrial tissue when taking biopsies. Consistently, the differences in level of known atrial-enriched proteins between sinus node tissue and atrial tissue is very small. Recent RNAseq gene expression profiling (Vedantham 2015) of purified mouse sinus node regions and atrial tissue revealed much greater differences in expression of Cx43, Cx40, Nppa, Nppb, Kcnj3, etc than reported here. Interestingly, sinus node enrichment of Hcn4 is similar between the published RNA profiles and proteomic analysis presented here (>100 fold) suggesting the discrepancies do not result from mRNA vs protein quantification. This needs to be addressed as the current quantifications may be biased to sinus nodal tissue-derived protein enrichment.

It is important to stress that a strong correlation between mRNA profiles and proteome profiles cannot be assumed. This is a fact. As stated in a leading edge review in Cell by Aebersold and colleagues¹⁷ in 2016: “there is no trivial relationship between the concentration of a transcript and the concentration(s) of the protein(s) derived from a particular locus”. In 2011, Schwanhäusser *et al.*¹⁶ investigated the relationship between mRNA and protein levels using RNA-seq and absolute quantification strategies based on pSILAC. They found that proteins on average were 2,800 times more abundant than corresponding transcripts and that mRNA and protein correlated with a correlation coefficient of $R^2 = 0.41$. Thus, a direct comparison between protein and mRNA abundances may not be appropriate.

Concerning our approach for isolating the sinus node, please also see our detailed responses to Reviewer #2 Question 2, as well as Reviewer #2 additional Comments 2.

To follow up on the points raised by the reviewer we did compare our reported proteome data to mRNA sequencing data reported by Vedantham *et al.*⁷, who used laser microdissection in a reporter mouse strain to accurately dissect SN tissue and adjacent RA tissue. This comparison needs to be evaluated in light of the concerns we express above in addition to the differences in mice studied: Vedantham’s RNAseq data is based on 14 days postnatal mural hearts, P14,

whereas our proteome data is based on 12 weeks old mice from a different strain. For all quantified proteins in our study, we retrieved the corresponding counts and expression differences on RNA level from Vedantham *et al.*, and compared the two values for each protein. Protein and mRNA are indeed correlated, but only with a medium correlation (Pearson correlation coefficient $R=0.4$), as shown in panel a of the figure below.

Comparison of SN vs RA expression differences in mRNA and proteome data. **a.** Correlation of protein intensity from our study to mRNA counts from Vedantham *et al.* Color gradient reflects point density (light blue = highest point density). **b.** Boxplots of difference distributions of proteome and mRNA data show wider spread of mRNA distribution, but similarity in central quantiles. **c.** Quantile-quantile plot comparing both distributions shows divergence mainly in the most extreme values, while central part of the distributions are similar. **d.** Two-dimensional density distribution of mRNA and protein ratios shows centering of the distribution at zero (light blue = highest point density). “Expr. Diff.” always indicates expression difference of SN / RA.

Overall the RNAseq dataset showed a slightly wider distribution than the proteome dataset (panel b), i. e. indeed some mRNA ratios between the tissues take more extreme values than protein ratios. However, this difference is mainly caused by outlier values while the central quantiles of the distribution do not differ substantially (panel c). To test whether protein ratios

deviate substantially from RNA ratios, we then investigated the difference between mRNA and protein ratio for each gene (panel d). We found that the distribution was centered at zero, i. e. most genes showed similar ratio at protein and mRNA level. The reviewer expresses a concern that contamination of SN samples by RA tissue may globally underestimate ratios for atrial-enriched proteins, which would bias the dataset towards sinus nodal tissue-derived protein enrichment. If such a bias were present, the comparison of ratios (panel c) would show a skew where mRNA ratios were systematically greater or smaller than protein ratios. In this context, the reviewer further mentions compressed ratios for known atrial-enriched proteins as compared to mRNA ratios (Cx43, Cx40, Nppa, Nppb, Kcnj3). However when assessing ratios for all proteins significantly enriched in RA in our study (that could be matched to corresponding mRNA ratios), we find that 175 show higher protein ratio in favour of RA than mRNA ratio, while only 59 show higher mRNA than protein ratio. That is, while the selected examples did show higher ratios on mRNA level, the majority of significant proteins shows the opposite behaviour with higher measured protein ratios than mRNA ratios.

Although the mRNA and proteome datasets showed the expected low correlation, a few proteins stood out with consistent tissue enrichment in both datasets. In SN, these were Hcn4, Hmcn2, Cma1, Vsn1, Lmod1, Cntn2, Acta2, Myh11 and Hcn1, as shown in the figure below. Consistent enrichment in RA in both datasets was found most prominently for Bmp10, Nppa, Nppb, and Comp. Given the difference in mouse age and strain, these proteins should be further scrutinized to determine whether the results are biologically meaningful.

We do agree with the reviewer that it in certain contexts may be useful to have access to comparisons of mRNA and proteome data. Accordingly we have added a new Supplementary Figure S3, on page 3 of the manuscript:

“We find the transcriptomics and proteomics data to be medium correlated (Supplementary Fig. S3), which is expected, and supports the notion of complementarity of the two sets of information^{16, 17}.”

On page 8 of the manuscript:

“The sinus node has previously been investigated at the transcriptome level⁷.”

3) The quantification should be compared to the various RNA-seq/expression profiling data sets of sinus nodal and atrial tissues. Many of the proteins or level differences reported here as novel can be found in these data sets. Moreover, mRNA and protein level comparisons may provide insight into discrepancies, which then can be tested.

As explained above, a direct comparison between protein and mRNA abundances may not be appropriate. We have included the comparisons that can be made, as outlined in the response to your previous question.

4) While the computer modeling is not within my field of expertise, I have the impression that it does not provide novel insights. The key currents of the sinus node and their differences with those of the atrial working myocytes are well established, and the model recapitulates that. Interventions to address the role of the different components and mechanisms would be required to gain novel insight. Several of these interventions (e.g. Hcn4 knock down) have already been published. Moreover, extrapolating channel activity from protein abundance ignores important post-translational modifications, localization, etc. involved in channel activity and currents.

The computer modelling is important for the following reason: in this study, we show that calcium clock proteins are not differentially expressed between sinus node and atrial muscle, whereas membrane clock proteins are; this suggests that the membrane clock and not the calcium clock is important for pacemaking; consistent with this, the computer modelling shows that the differences in membrane clock proteins can account for pacemaking. The modelling is essential for this line of reasoning. Without the modelling, we would not know whether, theoretically at least, the differences could account for pacemaking. And without our proteomics data we could not have known what the relative contributions of the different currents are. This aspect does not come from the modeling, but from our proteomics data.

The modelling is also an important proof of principle. We show for the first time that protein expression can be used in computer model development of action potentials. Of course, for the reasons you mention (localisation and post-translational modification), the modelling will not be precise, but nevertheless it is an important analytical tool that can be used to predict the possible consequences of a particular pattern of expression of ion channel proteins.

Reviewer #4 (Remarks to the Author):

I only have minor comments related to the modeling. The choice of model (Shen et al) should be justified. There are well-established alternatives, why was this model used?

The model of Shen *et al.* is the only model of the mouse atrial action potential. There are no other models available. We now state this on page 4 of the paper.

The gain of automaticity after changing the current density to sinus node expression levels is certainly an argument for the membrane-clock. It would strengthen the argument if one could also show the opposite effect, i.e. modifying a model of the sinus node to atrial myocyte expression levels would induce a loss of automaticity.

As requested by the reviewer, we have now done this. We used the model of Kharche et al.¹⁸ of the mouse sinus node cell action potential as a starting point and were able to successfully convert the sinus node action potential to an atrial-like action potential using our proteomics measurements of ion channels. The model was modified to include discrete ionic currents carried by the different HCN channels. The model was then modified in the same way as before, but in reverse. In addition, because the mass spectrometry provided no data on Kir2 channels (ion channels thought to be responsible for $I_{K,1}$), $I_{K,1}$ from the model of the mouse atrial cell action potential was incorporated. This resulted in the successful conversion of the sinus node action potential to an atrial-like action potential (non-spontaneous, triangular action potential shape).

We have added a description of this on pages 4 of the paper:

“We also did the reverse. In an analogous manner we modified a biophysically-detailed model of a mouse sinus node cell developed by Kharche *et al.*¹⁸. We were able to successfully convert the sinus node action potential to an atrial-like action potential (non-spontaneous, triangular action potential shape; see Supplementary Information for further details).”

We included further details in the Supplementary Material on page 6 and 7:

“We also did the reverse. We used the model of Kharche *et al.*¹⁸ of the mouse sinus node cell action potential as a starting point. The model was modified to include discrete ionic currents carried by the different HCN channels. The model was then modified in the same way as above, but in reverse. To successfully convert from the sinus node action potential to an atrial-like action potential (non-spontaneous, triangular action potential shape), we had in addition to make a change in the background inward rectifier K⁺ current ($I_{K,1}$). Because the mass spectrometry provided no data on Kir2 channels (ion channels thought to be responsible for $I_{K,1}$), $I_{K,1}$ from the model of the mouse atrial cell action potential was incorporated. This resulted in the successful conversion of the sinus node action potential to an atrial-like action potential. Given the lack of proteomic evidence for Kir2, one could speculate whether it is possible that not Kir2 channels are responsible for $I_{K,1}$, but instead TASK1 and/or Kir3 channels may be responsible which do change in the expected manner if they are responsible for $I_{K,1}$.”

We have not illustrated the results in the manuscript but the results are shown below for the benefit of the reviewer.

References

1. Dobrzynski, H. et al. Structure, function and clinical relevance of the cardiac conduction system, including the atrioventricular ring and outflow tract tissues. *Pharmacology & Therapeutics* 139, 260-288 (2013).
2. Dobrzynski, H., Boyett, M. R. & Anderson, R. H. New Insights Into Pacemaker Activity. *Circulation* 115, 1921-1932 (2007).
3. Verkerk, A. O. et al. Pacemaker current (I_f) in the human sinoatrial node. *European Heart Journal* 28, 2472-2478 (2007).
4. Chandler, N. J. et al. Molecular architecture of the human sinus node - insights into the function of the cardiac pacemaker. *Circulation* 119, 1562-1575 (2009).
5. Tellez, J. O. et al. Differential expression of ion channel transcripts in atrial muscle and sinoatrial node in rabbit. *Circulation Research* 99, 1384-93 (2006).
6. Doll, S. et al. Region and cell-type resolved quantitative proteomic map of the human heart. *Nature Communications* 8, 1469 (2017).
7. Vedantham, V., Galang, G., Evangelista, M., Deo, R. C. & Srivastava, D. RNA sequencing of mouse sinoatrial node reveals an upstream regulatory role for Islet-1 in cardiac pacemaker cells. *Circulation research* 116, 797-803 (2015).
8. Liu, J., Dobrzynski, H., Yanni, J., Boyett, M. R. & Lei, M. Organisation of the mouse sinoatrial node: structure and expression of HCN channels. *Cardiovascular Research* 73, 729-738 (2007).
9. Ayer-Lelievre, C., Dahl, D., Bjorklund, H. & Seiger, A. Neurofilament immunoreactivity in developing rat autonomic and sensory ganglia. *Int J Dev Neurosci* 3, 385-99 (1985).
10. Verhey, K. J., Dishinger, J. & Kee, H. L. Kinesin motors and primary cilia. *Biochem Soc Trans* 39, 1120-5 (2011).
11. Ishikawa, H., Thompson, J., Yates, J. R., 3rd & Marshall, W. F. Proteomic analysis of mammalian primary cilia. *Curr Biol* 22, 414-9 (2012).
12. Hulsmans, M. et al. Macrophages Facilitate Electrical Conduction in the Heart. *Cell* 169, 510-522 e20 (2017).
13. Hu, J. M. et al. CD163 as a marker of M2 macrophage, contribute to predicte aggressiveness and prognosis of Kazakh esophageal squamous cell carcinoma. *Oncotarget* 8, 21526-21538 (2017).
14. Mitrofanova, L. B., Gorshkov, A. N., Konovalov, P. V. & Krylova, J. S. Telocytes in the human sinoatrial node. *J Cell Mol Med* 22, 521-532 (2018).
15. Lakatta, E. G. & DiFrancesco, D. What keeps us ticking: a funny current, a calcium clock, or both? *J Mol Cell Cardiol* 47, 157-70 (2009).
16. Schwanhaeusser, B. et al. Global quantification of mammalian gene expression control. *Nature* 473, 337 (2011).
17. Liu, Y., Beyer, A. & Aebersold, R. On the Dependency of Cellular Protein Levels on mRNA Abundance. *Cell* 165, 535-550 (2016).
18. Kharche, S., Yu, J., Lei, M. & Zhang, H. A mathematical model of action potentials of mouse sinoatrial node cells with molecular bases. *American Journal of Physiology* 301 (2011).

REVIEWERS' COMMENTS:

Reviewer #2 (Remarks to the Author):

The authors have carefully approached my comments in their revised submission. In my opinion they have satisfactorily addressed all of my concerns. In particular, they have added significant new data where appropriate, and provided detailed clarification in other instances. The single cell RNASeq was a nice addition and appreciated.

This manuscript is now ready for publication.

Reviewer #3 (Remarks to the Author):

The authors provide a satisfactory response. The included complementary sn-RNAseq data strengthens the manuscript. I have no further comments.

note: recently, RNAseq data of purified mouse pacemaker cardiomyocytes (and atrial cells) and of fetal human sinus node regions and atria was published (van Eif et al. Development 2019), which may be appropriate to mention in the current manuscript.